# PointDMM: A Deep-Learning-Based Semantic Segmentation Method for Point Clouds in Complex Forest Environments

**Jiang Li** , **Jinhao Liu * and Qingqing Huang**

School of Engineer, Beijing Forestry University, Beijing 100086, China; jiangge@bjfu.edu.cn (J.L.);
Huangqingqing@bjfu.edu.cn (Q.H.)
*  Correspondence: liujinhao@bjfu.edu.cn

**Abstract: Background.** With the advancement of "digital forestry" and "intelligent forestry", point cloud data have emerged as a powerful tool for accurately capturing three-dimensional forest scenes. It enables the creation and presentation of digital forest systems, facilitates the monitoring of dynamic changes such as forest growth and logging processes, and facilitates the evaluation of forest resource fluctuations. However, forestry point cloud data are characterized by its large volume and the need for time-consuming and labor-intensive manual processing. Deep learning, with its exceptional learning capabilities, holds tremendous potential for processing forestry environment point cloud data. This potential is attributed to the availability of accurately annotated forestry point cloud data and the development of deep learning models specifically designed for forestry applications. Nonetheless, in practical scenarios, conventional direct annotation methods prove to be inefficient and time-consuming due to the complex terrain, dense foliage occlusion, and uneven sparsity of forestry point clouds. Furthermore, directly applying deep learning frameworks to forestry point clouds results in subpar accuracy and performance due to the large size, occlusion, sparsity, and unstructured nature of these scenes. Therefore, the proposal of accurately annotated forestry point cloud datasets and the establishment of semantic segmentation methods tailored for forestry environments hold paramount importance. **Methods.** A point cloud data annotation method based on single-tree positioning to enhance annotation efficiency was proposed and challenges such as occlusions and sparse distribution in forestry environments were addressed. This method facilitated the construction of a forestry point cloud semantic segmentation dataset, consisting of 1259 scenes and 214.4 billion points, encompassing four distinct categories. The pointDMM framework was introduced, a semantic segmentation framework specifically designed for forestry point clouds. The proposed method first integrates tree features using the DMM module and constructs key segmentation graphs utilizing energy segmentation functions. Subsequently, the cutpursuit algorithm is employed to solve the graph and achieve the pre-segmentation of semantics. The locally extracted forestry point cloud features from the pre-segmentation are comprehensively inputted into the network. Feature fusion is performed using the MLP method of multi-layer features, and ultimately, the point cloud is segmented using the lightweight PointNet. **Result.** Remarkable segmentation results are demonstrated on the DMM dataset, achieving an accuracy rate of 93% on a large-scale forest environment point cloud dataset known as DMM-3. Compared to other algorithms, the proposed method improves the accuracy of standing tree recognition by 21%. This method exhibits significant advantages in extracting feature information from artificially planted forest point clouds obtained from TLS. It establishes a solid foundation for the automation, intelligence, and informatization of forestry, thereby possessing substantial scientific significance.

**Keywords:** forest point cloud; 3D laser; deep learning; semantic segmentation



## 1. Introduction

The current forest area in China is 3.46 billion mu, ranking fifth in the world. The forest stock volume is 19.493 billion cubic meters, ranking sixth in the world. The area of

artificial forests is 1.314 billion mu, ranking first in the world. The total carbon storage of forests and grasslands reaches 11.443 billion tons, ranking among the top in the world. The key to the intelligent development of forestry lies in how to quickly and accurately obtain three-dimensional and rich information about forest areas and tree structures and achieve precise mapping and monitoring of forest resources. Traditional manual surveys and measurement methods are inefficient and cannot meet the modern forestry's demand for large-scale and efficient acquisition of three-dimensional structural data.

To replace time-consuming and labor-intensive manual surveys with poor accuracy, commonly used methods for forest environmental surveys include hyperspectral, image, and point cloud. LiDAR-derived point cloud data record the three-dimensional shape information of targets. By conducting non-contact remote scanning of the entire forest area, high-density three-dimensional point cloud data of individual trees and even the entire forest area can be obtained, which fully captures the three-dimensional information of tree crowns, trunk shapes, branch structures, and more. Point cloud data can accurately reflect the three-dimensional scenes of forests and be used to construct and present digital forest systems. These rich three-dimensional point cloud data can be used to build detailed digital forest structure models, facilitating the extraction of parameters such as tree species, tree height, diameter at breast height, crown width, etc., from point clouds. This enables accurate measurement and dynamic monitoring of the spatial distribution and growth status of trees, greatly improving the accuracy of understanding forest resources in the area. Furthermore, point cloud data from different periods can be compared to monitor dynamic changes such as forest growth and logging processes and assess the increase or decrease in forest resources. Additionally, using point cloud data to construct detailed digital forests can simulate various forestry operations, such as transportation planning and logging operations, thereby improving the efficiency of forestry operations. In summary, compared to manual measurement methods, three-dimensional LiDAR scanning can cover a larger range and obtain more comprehensive three-dimensional structural data. It is not limited by terrain and environment, making it more efficient and convenient to operate.

Since its inception in the 1960s, laser technology has undergone significant advancements, particularly in terms of measurement accuracy. Consequently, its applications have proliferated across diverse sectors, including the military, industrial manufacturing, civil engineering, agriculture, and forestry. Within the realm of forestry, airborne light detection and ranging (LiDAR) technology has reached a state of maturity, having been employed as early as the 1980s. Pioneering work by Nelson, Ross [1], and colleagues demonstrated the utility of airborne LiDAR for measuring vertical forest features, such as tree height and ground distance. Their findings indicated that the margin of error for tree height measurements was less than one meter when compared to photogrammetric techniques. Schreier [2] utilized airborne LiDAR technology to scan forested areas and demonstrating that laser-generated point clouds could precisely differentiate between the ground and various types of vegetation. Furthermore, the technology is capable of distinguishing between coniferous and broadleaf forests based on metrics such as distance information, reflectance, and other parameters. Since the 1990s, airborne LiDAR technology has evolved, garnering increasing interest from researchers in forestry. The technology has been deployed to obtain comprehensive data on various aspects of forests, including growth factors, ecological conditions, vertical structure, and biomass over expansive areas. In contrast to airborne and satellite-based LiDAR systems, vehicle-mounted LiDAR offers the advantage of capturing more granular data on forest stand structures, owing to its high-density and high-precision point cloud capabilities. In recent years, research efforts have increasingly pivoted towards high-precision forest modeling and targeted identification. For example, Merlijn Simonse et al. explored the utility of vehicle-mounted LiDAR for forest resource surveys, extracting key parameters like stand positions and diameters at breast height from 3D point cloud data, with a particular emphasis on data-processing methodologies [3]. Initially, the researchers employed Z-coordinates to identify the lowest points across various horizontal planes, thereby establishing a digital ground model. Subsequently, they applied the Hough

transform technique to filter the point cloud data, allowing them to accurately pinpoint stumpage positions and their respective diameters at breast height. In conclusion, the study offers prospective insights into the broader applications of LiDAR technology for comprehensive extraction of stumpage parameters, such as tree species identification, tree height, canopy area, and wood defects, among others. Chris Hopkinson [4] employed a synergistic approach, utilizing both ground-based and airborne LiDAR systems to scan tree canopies. Subsequent calibration calculations were executed with a high level of accuracy, enabling the derivation of both leaf area index and leaf profile models within a 1 km range. In a separate study, Aleksey Golovinskiy [5] focused on target identification in urban settings by employing a shape-feature approach. The researchers initially captured 3D point cloud data within the urban landscape and employed clustering techniques for point cloud segmentation. This enabled the differentiation of foreground and background entities. Following this, shape features were extracted, and labeled data were used for training. The final step involved the application of support vector machines for robust target recognition. In Germany, Bienert et al. [6] utilized vehicle-mounted 3D LiDAR technology to estimate the volume of trees. Initially, a rudimentary method was employed in which the point cloud was conceptualized as a stereo pixel structure. Tree volume was then calculated based on the number of filled stereo pixels. However, this approach led to overestimation issues. Consequently, refinements were made to address these inaccuracies, resulting in a more reliable method. In another study, Fabrice Monnier et al. [7] explored the identification of urban street trees using 3D point clouds. The team defined various tree features, including volume, linear, cylindrical, and planar characteristics. Each feature was independently recognized. A probabilistic relaxation method was then implemented to filter out noise and insignificant point cloud structures. The aggregated data indicated that, even in complex urban settings, trees could be effectively identified using individual features. The aggregated data corroborated that in complex urban landscapes, tree identification could be effectively accomplished using individual features. In a separate study, Rutzinger et al. [8] focused on tree recognition using 3D point clouds and constructed three-dimensional models for standing trees. They initially employed clustering algorithms to segment the point cloud data, isolating points that were situated at least 0.5 m above the ground. Subsequently, point density for each cluster was calculated. It was observed that the density of point clouds in the tree crown was substantially lower than in other areas, enabling effective tree recognition. The images in the manuscript reveal that the experiments were conducted on an urban road with sparse tree coverage, achieving an 85% recognition rate under these conditions. Hyyti et al. [9] employed a 2D laser scanner, rotated it around a baseline to enable 3D scanning, and generated 3D point cloud data of a forest area. Utilizing circular arc features, they successfully identified tree trunks and estimated their positions. However, their method exhibited lower accuracy in calculating trunk diameters, resulting in errors of less than 4 cm within an 8 m scanning radius. In a similar vein, Pyare Pueschel et al. [10] utilized a FARO 3D laser scanner to scan a forest area, from which they extracted tree locations, diameters, and wood volumes. Their study also evaluated the impact of varying scanning modes and curve-fitting methods on the accuracy of diameter and volume calculations, taking into account the effect of trunk occlusion. Sandeep Gupta et al. [11] focused on 3D modeling of the vertical structure of tree canopies and individual trees. Utilizing airborne LiDAR, they acquired point cloud data, performed statistical analyses on the height distribution, and stored the data in an octree structure for further processing. Yangyan Li et al. [12] extended point cloud studies to the domain of plant growth analysis. By capturing 3D point cloud data over an extended time frame—referred to as 4D point clouds—they examined plant growth patterns. Although numerous methods currently exist for extracting forest parameters from laser point clouds, both domestically and internationally, they encounter challenges that hinder the broader application of LiDAR in practical forestry management.

Due to the complexity of the forest environment and the large amount of point cloud data, traditional post-processing methods require manual intervention and multiple steps,

which cannot serve as the technical foundation for real-time operations and rapid surveys. Deep learning, with its powerful learning ability, has great potential in handling forest environment point clouds. Currently, deep learning has five development directions in point cloud recognition:

**Volumetric-based methods:** typically involve voxelizing point clouds into 3D grids and applying 3D convolutional neural networks. However, the computational requirements and sparsity of stereo data after rasterization hinder the development of this approach. Some proposed solutions, such as CNNS [13], FPNN [14], and Vote3D [15], have attempted to address these challenges, but difficulties still arise when dealing with large amounts of point cloud data.

**Multi-view CNNs methods:** attempt to transform 3D point clouds or shapes into 2D images and utilize 2D convolutional networks for classification. While this method achieved good recognition results at the time, it is difficult to extend to large scenes and 3D tasks, such as point cloud classification. Furthermore, 2D multi-view images only approximate 3D scenes and do not provide a true and lossless representation of the geometric structure, resulting in less ideal results in complex tasks.

**Pointwise MLP methods:** Utilize multiple shared multi-layer perceptrons to independently model each point and then aggregate global features using symmetric aggregation functions. These methods have made significant advancements in recent years. Point-Net [16], introduced by researchers from Stanford University in 2017, directly processes unordered point clouds as input data for recognition and semantic segmentation tasks. PointNet++ [17], an extension of PointNet, addresses the limitation of extracting local information by utilizing Farthest Point Sampling (FPS) and Multi-Scale Grouping (MSG). Other methods, such as POINTWEB [18] and PointSIFT [19], focus on extracting contextual features from the local neighborhood of point clouds and incorporating the concept of the SIFT algorithm for point convolution.

**3D convolutional kernels methods:** Compared to convolutional kernels defined on a two-dimensional grid structure, designing convolutional kernels for three-dimensional point clouds poses greater challenges due to their irregularity. Current three-dimensional convolution methods can be categorized into continuous convolution and discrete convolution, depending on the type of convolutional kernel used. PointCNN [20] addresses the difficulty of applying convolutional operations to irregular and unordered point cloud data by employing point convolution. Flownet3D [21] extracts features and computes their correlation. Other methods, such as Spherical Convolutional Spectral CNN [22], KPCONV [23], and PointConv [24], propose novel techniques for modeling the geometric relationship between neighboring points and performing convolution operations.

**Graph-based methods:** SuperPointGraph [25], GCNN [26]-utilize graph convolution for the effective processing of point clouds. ClusterNet [27] generates rotation-invariant features and constructs a hierarchical structure of point clouds using an unsupervised approach.

The advancement of deep learning methodologies has shown promise in improving the efficiency of topographic LiDAR technology for forestry applications. These methodologies offer various approaches for point cloud recognition, each with its own strengths and limitations. Further research and development in this field will contribute to the integration of laser measurement technologies into operational forestry practices. However, in practical applications, traditional direct annotation methods are time-consuming and inefficient due to the complex terrain, foliage occlusion, and uneven sparsity of forestry point clouds. Meanwhile, due to the unstructured characteristics of the vast forestry point cloud scenes, such as occlusion and sparsity, directly applying deep learning frameworks to forestry point clouds results in low accuracy and poor performance. Therefore, a precisely annotated forestry point cloud dataset is proposed, and a point cloud semantic segmentation method suitable for forestry environments is established, which is of great significance.

To address these challenges and develop a more effective training model tailored to forestry scenarios, we conducted an in-depth investigation of forest landscapes. Leveraging

multi-sensor fusion LiDAR technology, we collected high-quality data and performed precise semantic annotations, resulting in the creation of the "DMM" dataset.

Recognizing that forestry point cloud data are often highly obscured and present other issues, we engineered a feature extraction DMM module. This module is specifically designed to optimize the extraction of features from forestry point clouds. Additionally, we developed a semantic geometric segmentation algorithm that categorizes point clouds based on shared features. As a result, we propose an end-to-end point cloud processing framework called pointDMM.

## 2. Methods and Materials

In this section, we outline the annotation framework of our data acquisition system and introduce the DMM dataset for forestry scenes. Additionally, we discuss the DMM module, which is specifically designed for pre-segmenting multi-feature point clouds, and present pointDMM, a method for segmenting depth point clouds in garden scenes.

### 2.1. Study Area

Data collection was conducted at two distinct locations. As shown in Figure 1, the first location, Gao Yang County Forestry District, is situated at coordinates 115°38′ E and 38°37′ N. It experiences an average annual rainfall of 515.2 mm. The second location, Beijing Dongsheng Bajia Park, is located at No. 5 Shuangqing Road, Dongsheng Township, Haidian District, Beijing, China. Its coordinates are N: 40°01′4.78″ and E: 116°20′40.63″. This park, the largest of its kind in Beijing, is located within the temperate monsoon zone. It has an average annual rainfall of 688.26 mm and an average annual temperature of 13.1 °C. The park spans approximately 615.83 hectares and boasts a rich diversity of plant species, including 21,700 trees with significant crown and flower coverage. The green area coverage exceeds 90%. In this study, we focus on a plantation forest spanning about 20 mu, primarily composed of Tsubaki and Populus species.

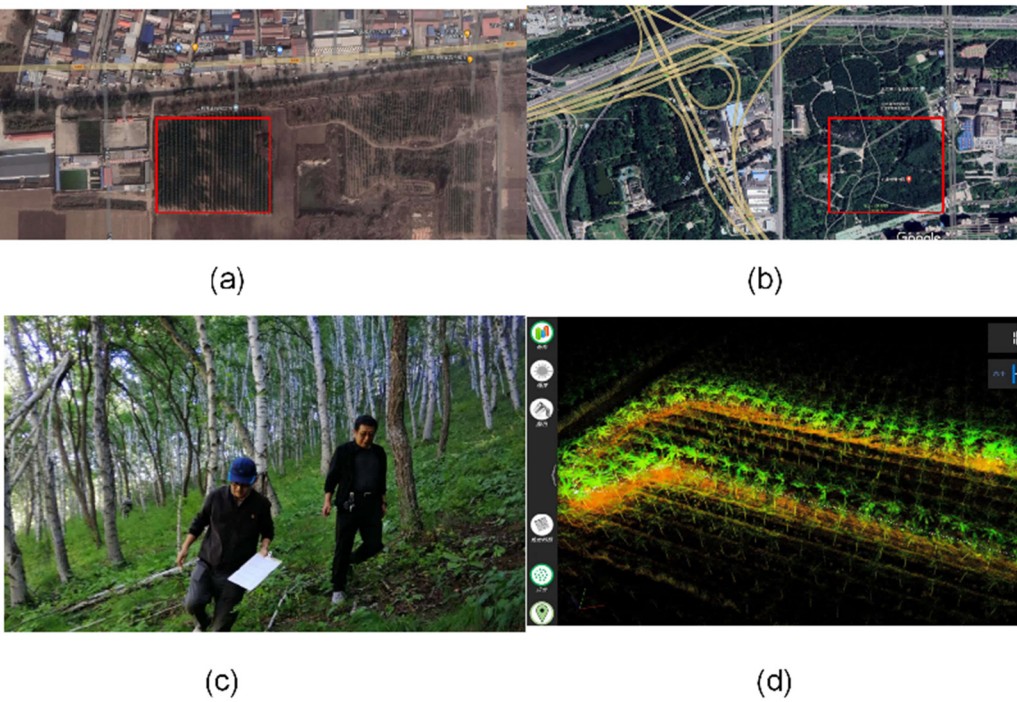

**Figure 1.** Study area: The study area consists of four data collection sites, as shown in Figure (**a**–**d**). Figure (**a**) represents the data collection site located in the forest area of Gaoyang City, Hebei Province. Figure (**b**) depicts the data collection site situated in the Bajia homestead in Haidian District, Beijing. Figure (**c**) displays the data collection site within the forest sample area. Lastly, Figure (**d**) showcases the real-time display of data collection, with the white color indicating the walking track.

*2.2. Data Collection Platform*

We begin this section by providing details of the laser scanner used for collecting point cloud data. As shown in Figure 2, Our backpack-type acquisition system utilizes the RS-LiDAR-16, developed by Shenzhen Suteng Juchuang Technology Co., Ltd. (Shenzhen, China). This state-of-the-art LiDAR unit is designed for applications in autonomous vehicle environment perception, robotics, and UAV mapping. The RS-LiDAR-16 employs a hybrid solid-state LiDAR approach, integrating 16 laser transceiver components capable of measuring distances up to 150 m with an accuracy of $\pm 2$ cm. The unit produces up to 300,000 data points per second, offering a horizontal measuring angle of 360° and a vertical angle ranging from $-15°$ to 15°. The device performs exceptionally well under adverse visibility conditions such as sandstorms, haze, rain, or dense vegetation, thanks to RIEGL's unique LiDAR technology. During our fieldwork, we collected a total of 197 point clouds over an area of 8.1 hectares. These point clouds were stored in LAS1.4 format. Given the large amount of data, I have organized all the experimental results mentioned in this document. The point cloud of the intermediate steps in the experiment has a total size of 8.8 GB and has been uploaded to Quark Cloud Drive. Please see https://pan.quark.cn/s/fdce3d6aedac (accessed on 30 October 2023) (Figure 2).

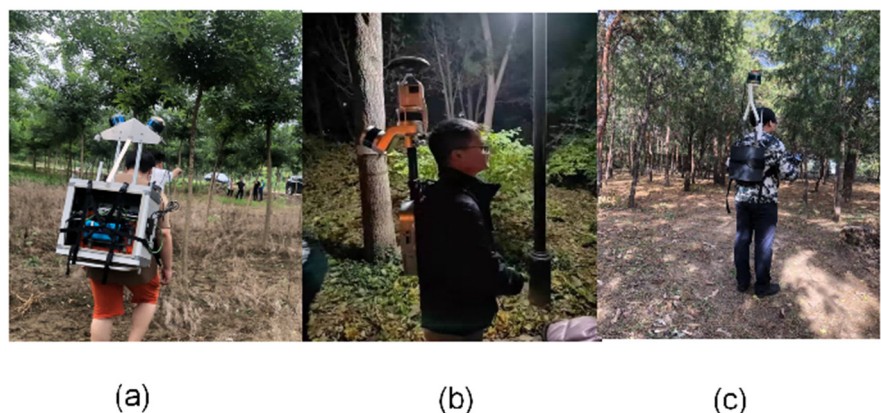

**Figure 2.** Data acquisition Figure (**a**–**c**) are our backpack LIDAR, where Figure (**a**) is a double-headed LIDAR, Figure (**b**)'s double-headed LIDAR one collects horizontal direction and one vertical direction, and Figure (**c**) is a single-headed LIDAR.

The SLAM technology of LiDAR has made significant progress in recent years, both in theoretical research and practical applications. With the advancement of sensor technology and computing power, we have adopted SLAM technology for point cloud collection in forestry environments. SLAM technology can work stably in complex and changing forest environments without the need for prior environmental information or GPS signals. It offers cost-effectiveness compared to traditional aerial or satellite remote sensing as SLAM-based ground point cloud data collection has lower costs and can be updated more frequently. Additionally, SLAM systems can be installed on various mobile platforms, such as drones, mobile robots, or handheld devices, providing flexible data collection solutions for different forestry applications. Furthermore, SLAM technology allows researchers and forestry workers to view point cloud data in real time, enabling them to make timely decisions and adjustments. Minimal environmental impact: Compared to traditional forestry measurement methods, SLAM-based point cloud data collection has a lower impact on the environment and is more environmentally friendly. Furthermore, SLAM technology enables data collection at various scales, ranging from individual trees to entire forests, providing abundant resources for forestry research and management. The application of SLAM technology in forestry point cloud data collection presents both new opportunities and challenges for forestry research and management. In this article, the Roboscene lidar-fusion inertial navigation backpack SLAM collection system and the Lioslam [28] algorithm are utilized for data collection. By tightly coupling the lidar and inertial measurement unit



(IMU) odometry system, the system achieves high-precision, real-time trajectory estimation, and map construction through smoothing and mapping techniques. The system builds a laser inertial odometry based on a factor graph, allowing the integration of relative and absolute measurement data from multiple sources, including loop closure detection. To remove distortion from point clouds, the system utilizes pre-integrated data from the IMU and provides initial estimation for laser odometry optimization. Through scan matching, selective keyframe introduction, and an efficient sliding window strategy within a local range, LIO-SLAM ensures real-time performance in forestry applications while maintaining high-precision trajectory estimation and map construction capabilities in forest environments.

The implementation of the Lioslam algorithm consists of several steps. Firstly, pre-processing is performed to correct the distortion of the input point cloud using IMU data. The point cloud data are then segmented into ground and non-ground points. Next, visual odometry is estimated by aligning the current frame with the previous frame's point cloud data to determine the relative transformation between the two frames. The initial estimate provided by the IMU data assists in this alignment process. Subsequently, factor graph optimization is conducted by adding the laser odometry and IMU pre-integration data as factors to the factor graph. When loop closure is detected, loop closure constraints are added to the factor graph. Optimization algorithms, such as GTSAM or g2o, are employed to optimize the factor graph and obtain a globally consistent trajectory and map. The fused point clouds are then combined with the optimized trajectory to construct a global map. Optionally, map sparsification or downsampling can be performed to enhance efficiency. Loop closure detection is achieved by utilizing point cloud descriptors (FPFH) to match the current frame with historical frames and identify loop closures. If a loop closure is detected, loop closure constraints are added to the factor graph. Finally, the optimized trajectory and 3D map are outputted, enabling further path planning or navigation.

Overall, the utilization of SLAM technology in forestry point cloud data collection offers numerous benefits, including minimal environmental impact, multi-scale data collection capabilities, and improved precision in trajectory estimation and map construction. The Lioslam algorithm, along with the ROboscene lidar-fusion inertial navigation backpack SLAM collection system, provides a comprehensive solution for efficient and accurate forestry research and management (As shown in Table 1).

To facilitate the training of deep learning models, we have organized the acquired data into three distinct, scaled datasets: DMM-1, DMM-2, and DMM-3. Each dataset is tailored to different scene scales, providing a comprehensive range of forestry environments. Specifically, DMM-1 focuses on individual trees, enabling accurate segmentation at a fine-grained level. DMM-2 emphasizes the semantic segmentation of local tree populations, allowing for a more detailed analysis of localized forestry environments. Finally, DMM-3 targets large-scale, multi-tree scenarios, enabling the assessment of expansive forest landscapes. For instance, DMM-1 serves as the ideal testing ground for single-tree segmentation. Meanwhile, DMM-2 and DMM-3 provide valuable insights into the semantic segmentation of localized forestry environments and expansive forest landscapes, respectively. Overall, these datasets offer a valuable resource for researchers and practitioners in the field of deep learning, facilitating the development and evaluation of advanced models for forestry analysis. The original point cloud can be seen Figure 3.

**Table 1.** RS16-line scanner equipment specifications and parameters.

| Training Hyperparameters | Parameter Values |
|---|---|
| Sensors | TOF method ranging 16 channels<br>Measurement: 40 cm to 150 m (20% target reflectivity)<br>Accuracy: ±2 cm<br>Angle of view (vertical): ±15° (30° total)<br>Angular resolution (vertical): 2° Viewing angle (horizontal): 360°<br>Angular resolution (horizontal/azimuth): 0.1° (5 Hz) to 0.4° (20 Hz)<br>Speed: 300/600/1200 rpm (5/10/20 Hz) |
| Laser | Class 1<br>Wavelength: 905 nm<br>Laser Emission Angle (full angle): 7.4 mrad horizontal, 1.4 mrad vertical<br>~300 k dots/s |
| Output | 100 Gigabit Ethernet<br>UDP packets contain<br>Distance information<br>16 line parameters<br>Rotation angle information<br>Calibrated reflectivity information |
| Mechanical/electronic operation | Power consumption: 12 w (typical)<br>Operating voltage: 9–32 VDC (requires interface box and stable power supply)<br>Weight: 0.87 kg (excluding data cable)<br>Operating voltage: 9–32 VDC (requires interface box and stable power supply)<br>Weight: 0.87 kg (excluding data cable)<br>Dimensions: Diameter 109 mm × Height 80.7 mm<br>Protection and safety level: IP67<br>Operating ambient temperature range: −30 °C~60 °C<br>Storage ambient temperature range: −40 °C~85 °C |

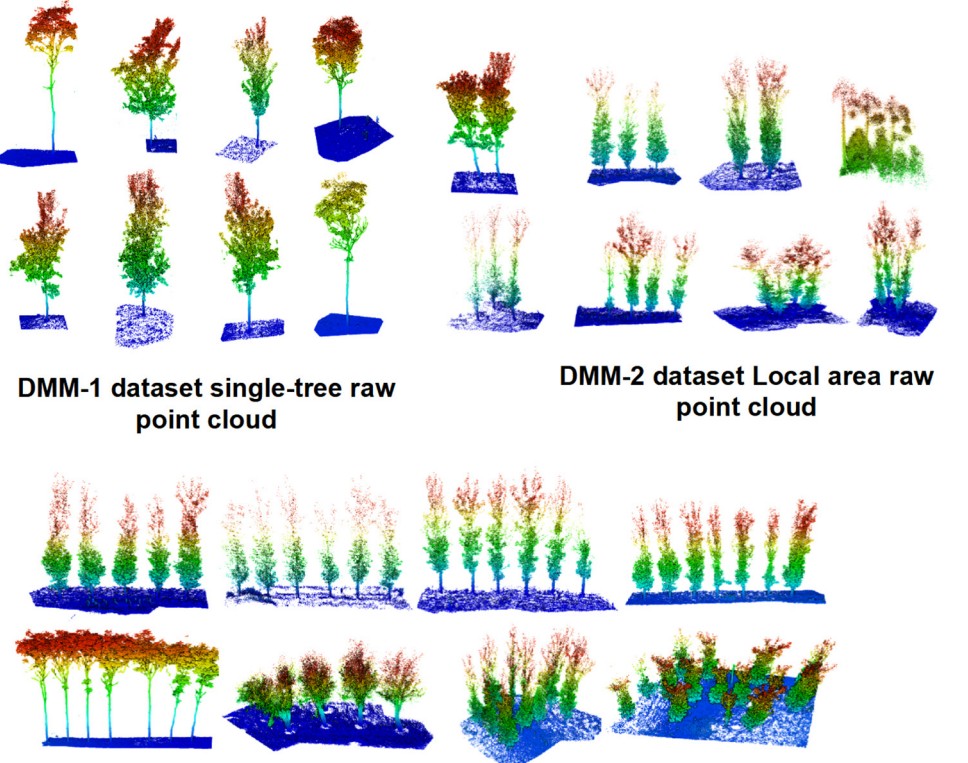

**DMM-1 dataset single-tree raw point cloud**

**DMM-2 dataset Local area raw point cloud**

**DMM-3 dataset Forest environment large scene raw point cloud**

**Figure 3.** The DMM dataset collected by backpack-style lidar includes the following: DMM-1 single tree point cloud dataset, DMM-2 forestry partial area point cloud dataset, and DMM-3 forest environment large scene point cloud dataset.

*2.3. Tree-Based Localization-Based Forestry Point Cloud Dataset Annotation Method*

There is currently a lack of an accurate annotated dataset for multi-forestry identification and segmentation based on the mobile information collection platform for the forestry environment. The forest rapid mobile measurement and collection platform faces difficulties in annotating large-scale, different density, and unstructured point clouds. The annotation process is inefficient and results in low accuracy. Additionally, there are challenges with sparse and occluded trees and other objects, as well as the undefined fractal structure, which can lead to annotation errors. The existing point cloud data for processing complex forestry environment information are insufficient, as it lacks data with severe occlusion, high density, complex terrain, multiple return information, and uneven scale. To address these issues, we propose a method for annotating forestry large-scale scene data based on single-tree positioning. Compared to commonly used outdoor datasets such as semantic3D, the forestry point cloud dataset has its own characteristics. Semantic3D is currently the largest and most popular static dataset, where each frame is measured from a fixed position using a ground-based LiDAR scanner. The main categories in this dataset are ground, vegetation, and buildings, with fewer moving objects. It includes 3D semantic scenes from rural and urban areas, with three distinct suburban categories. The proportions of each category also vary.

Due to the forestry environment, our dataset mainly consists of live standing trees without any buildings or pedestrian information. Therefore, our point cloud annotation method is based on single-tree localization and pre-segmentation. The method includes the following steps: loading point cloud labels, denoising and filtering normalization of the point cloud, calculating DBH (diameter at breast height) and CHM (canopy height model), pre-segmentation of individual trees in the point cloud, and fine annotation based on pre-segmentation. As shown in Figure 4,the specific steps are as follows:

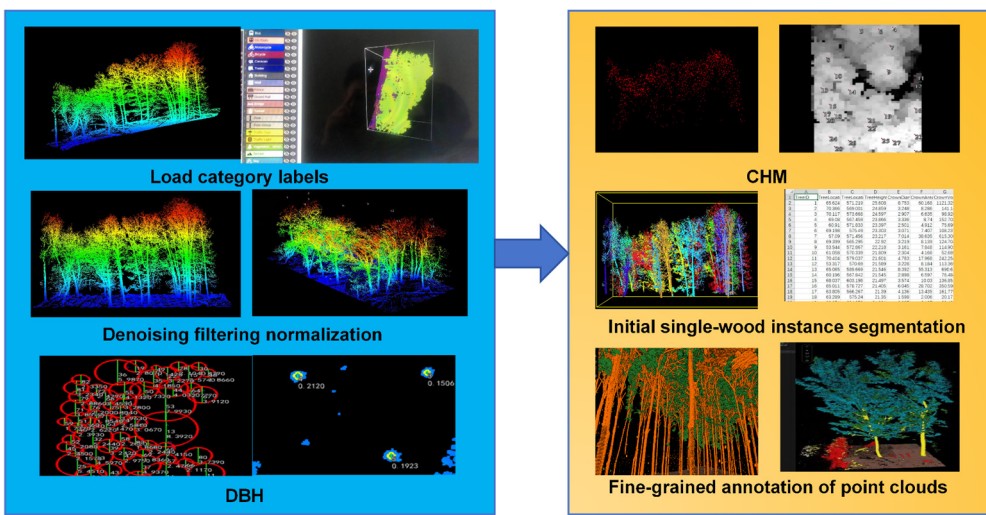

**Figure 4.** Based on single-tree pre-segmentation, the point cloud annotation method consists of several parts, including loading point cloud labels, point cloud denoising filtering normalization, calculating DBH, CHM, point cloud single-tree pre-segmentation, and fine annotation of point cloud based on pre-segmentation.

**1. Use the Semantic-Segmentation-Editor to load point cloud labels:** It is an open-source web-based semantic object annotation editor called Semantic Segmentation Editor, developed by Hitachi Automotive And Industry Lab. This tool is specifically designed for creating training data for machine learning semantic segmentation in the field of autonomous driving research and development. However, it can also be used to annotate other types of semantic object databases. It supports 3D point clouds generated by LIDAR (in .pcd format). This article provides instructions for installing meteor and configuring

the environment on Ubuntu 18.04. Its address is https://github.com/GerasymenkoS/semantic-segmentation-editor (accessed on 11 January 2023.).

**2. Denoising:** The denoising of point clouds is a critical preprocessing step aimed at eliminating noise and outliers from 3D point cloud data. Point cloud data often suffer from noise interference due to limitations in 3D scanning devices or imperfect image reconstruction techniques. The objective of denoising methods is to restore the true structure of the point cloud while preserving its essential features. This process involves analyzing each point in the point cloud to determine if it is noise and adjusting or removing it based on the attributes of its neighboring points. Efficient point cloud denoising not only enhances the accuracy of subsequent tasks, such as 3D reconstruction, classification, and recognition, but also enables the identification of outlier points based on the distribution of distances between each point and its neighbors. The denoising algorithm follows the following steps: Establish k-nearest neighbors for each point in the point cloud: for each point P in the point cloud, find its k nearest neighbors, where k is a predefined parameter; Calculate the average distance and standard deviation: for point P and its neighbors, calculate the average distance $\mu$ and standard deviation $\sigma$ from P to its neighbors; define a threshold: $T = \mu + \alpha * \sigma$, where $\alpha$ is a predefined coefficient used to control the strictness of denoising; remove outlier points: for point P and its neighboring points, if the distance from P to any neighbor is greater than the threshold T, consider P as an outlier point and remove it from the point cloud; iterative optimization: repeat the above steps until the number of outlier points in the point cloud is less than a predefined threshold or reaches the maximum iteration count; output the denoised point cloud. By following these steps, the denoising algorithm effectively removes noise and outlier points, resulting in a denoised point cloud that accurately represents the true structure of the data.

**3. Normalization**: The objective of point cloud normalization is to standardize the scale, position, or orientation of point cloud data with respect to a reference framework. Presented here is a step-by-step implementation of a straightforward point cloud normalization algorithm: Calculate the centroid: iterate through all points in the point cloud and compute the average x, y, and z coordinates to determine the centroid (x, y, z). Translate to the origin: for each point (x, y, z) in the point cloud, translate it such that the centroid is positioned at the origin; calculate the scale factor: compute the maximum distance, Dmax, from all points in the point cloud to the origin. Define a desired normalized radius, R. Calculate the scale factor as scale = R/Dmax; scale normalization: scale each point in the point cloud using the calculated scale factor; direction normalization: if necessary, ascertain the principal direction of the point cloud using methods such as principal component analysis (PCA). Rotate the point cloud to align its principal direction with a predefined direction, such as the z-axis; output the normalized point cloud; this algorithm initially relocates the centroid of the point cloud to the origin and subsequently scales it based on the maximum scale of the point cloud, ensuring that it falls within a standardized range. Additionally, if required, the direction of the point cloud can be adjusted. This normalization method establishes a unified reference framework for subsequent point cloud processing and analysis.

**4. DBH**: Diameter at breast height (DBH) refers to the diameter of a tree measured at a height of 1.3 m (or 4.5 feet) above the ground. This measurement is widely used in forestry and ecology to estimate the age, health, and growth rate of trees. DBH is a crucial parameter for assessing forest resources, calculating timber yield, and making informed forest management decisions. Typically, a tape measure or a specialized DBH measuring tape is utilized to obtain this value. By regularly measuring and recording the DBH of trees, researchers and forestry managers can monitor tree growth, health, and the overall condition of forest ecosystems. Single-tree segmentation based on seed points can be employed to derive parameters such as tree height, breast diameter, and crown diameter through the use of a single-tree segmentation algorithm. The breast diameter calculation method involves selecting point cloud data at the breast diameter position of an individual tree, fitting the DBH, and calculating the diameter of the fitted circle

to determine the breast diameter of the tree. The crown diameter can be obtained by measuring the crown area and using the area measurement formula $S = \pi r^2$ to calculate the crown diameter (2r). The following outlines the point cloud DBH estimation algorithm: Data preprocessing: apply a point cloud denoising algorithm to remove noise. Utilize a ground segmentation algorithm to separate ground points from non-ground points (such as trees and other objects). Locate breast height position: identify the highest point on the z-axis of the ground. Add 1.3 m to this height to determine the breast height position. Extract point cloud slice at breast height: take a small range (to evaluate based on point cloud density) above and below the breast height position and extract all points within this range to form a point cloud slice. Calculate the convex hull of the slice: employ a 2D convex hull algorithm to determine the convex contour of the tree trunk on the breast height slice. Calculate DBH: measure the maximum or average diameter of the convex hull, which will be the estimated DBH value. Optimization and calibration: if multiple trees or other objects interfere, clustering algorithms can be utilized to separate different objects and calculate DBH separately. Known reference objects can be used for scale calibration and to output accurate DBH values. In summary, DBH measurement at breast height plays a crucial role in assessing tree characteristics and making informed decisions in forest management. The point cloud DBH estimation algorithm provides a reliable method for accurately determining DBH values, contributing to the overall understanding and preservation of forest ecosystems.

**5. CHM**: CHM, short for canopy height model, is a two-dimensional data model that represents the height of the vegetation canopy on the ground. It is calculated by measuring the direct distance from the ground to the top of the vegetation. CHM is typically derived from remote sensing data, such as LiDAR or SAR, obtained from aerial or ground-based platforms. The CHM provides researchers with an intuitive way to observe and analyze the structure and height distribution of forests or other vegetation. It has wide applications in ecology, forestry, and environmental science, including biomass estimation, carbon storage, tree growth, and forest health monitoring. The process of generating a CHM involves several steps. The relative height for each point is calculated by subtracting the height of the corresponding DEM position from the Z value of the point. The canopy height model (CHM) is then generated by finding the point with the maximum relative height in each grid cell and assigning this maximum value to the corresponding grid cell in the CHM. smoothing techniques such as Gaussian filtering or other filtering methods can be applied. Finally, the CHM is outputted. Additionally, algorithm processing can be used to obtain data on the number, position, height, and crown width of individual trees. Overall, the CHM plays a crucial role in studying and understanding vegetation characteristics, and its applications are diverse in various scientific fields.

**6. Fine-grained annotation:** Pointly tree segmentation labeling. Leveraging state-of-the-art machine learning algorithms, the software effectively classifies and segments point cloud data, facilitating a quicker and more accurate understanding of the data. Pointly boasts an intuitive user interface that allows even those with limited experience in 3D data processing to quickly start. Furthermore, it supports various point cloud data formats and offers a wide range of data export and sharing functions, streamlining collaboration with team members and stakeholders. The resulting point cloud annotation can be seen in Figure 5, generating a comprehensive DMM dataset with labels for trees, low shrubs, land, and other categories.

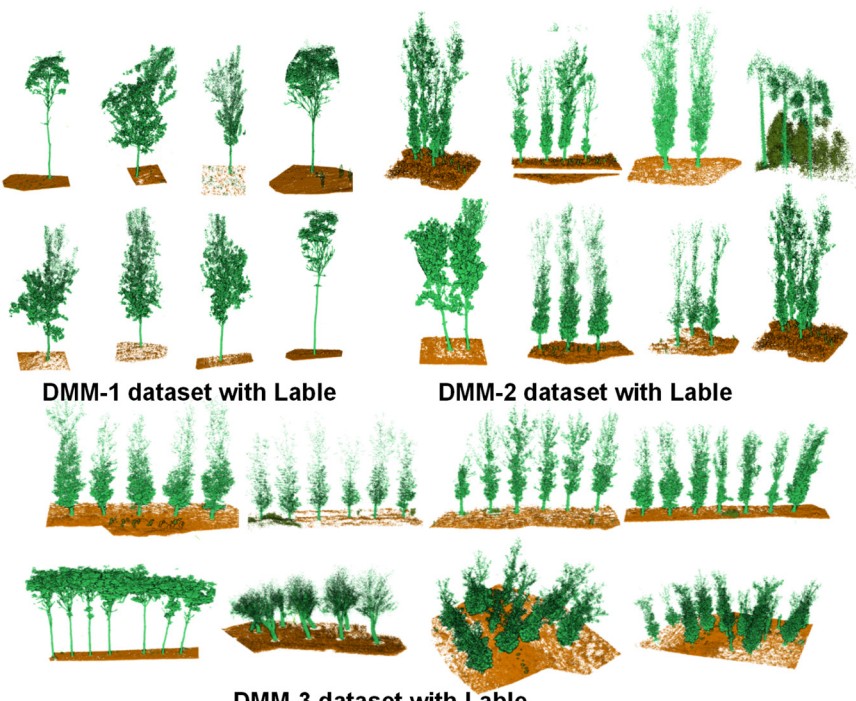

**Figure 5.** Semantic labels for DMM point cloud processing and labeling depict the sequential steps involved in the transformation of raw point cloud data into a comprehensive DMM dataset encompassing trees, low shrubs, land, and other distinct categories.

As shown in Table 2, SemanticKitti [29] introduces a large dataset to promote lidar-based semantic segmentation research. This dataset is annotated for all sequences of the KITTI Vision Odometry Benchmark, providing dense point annotations of the full field of view. Semantic3D [30] contains over 4 billion points and includes a variety of urban scenes, such as churches, streets, railway tracks, squares, villages, football fields, and castles. It provides detailed point cloud data scanned with state-of-the-art devices and includes eight category labels. In contrast, our dataset was collected using 16-line lidar in over 1259 scenes using a backpack-style laser LiDAR. It contains 2144 million points and is divided into tall trees, low shrubs, land, and other categories. Our dataset is further divided into three parts: DMM-1, DMM-2, and DMM-3. DMM-1 represents the point cloud of a single tree, and the original point cloud is shown in Figure 3, while the labeled label is shown in Figure 5. DMM-2 represents the point cloud of multiple trees in a small range, and the original point cloud is shown in Figure 3, with the labeled label shown in Figure 5. DMM-3 represents the point cloud of multiple trees in a large range, displayed by the cutting method, and the original point cloud is shown in Figure 3, with the labeled label shown in Figure 5.

**Table 2.** Comparison of different point cloud datasets.

| Dataset | Scenes | Points ($10^6$) | Classes | Sensor | Annotation |
| --- | --- | --- | --- | --- | --- |
| SemanticKITTI [29] | 23,201 | 4549 | 25 | Velodyne-64 | point-wise |
| Semantic3d [30] | 15/15 | 4009 | 8 | Ter-lidar | point-wise |
| Oakland3d [31] | 17 | 1.6 | 5 | SICk | point-wise |
| Freiburg [32] | 77 | 1.1 | 4 | SICK | point-wise |
| Wachtberg [33] | 5 | 0.4 | 5 | Velodyne 64 | point-wise |
| Paris-Lille-3D [34] | 3 | 143 | 9 | Velodyne-32 | point-wise |
| KITTI [35] | 7481 | 1799 | 3 | Velodyne-64 | bounding box |
| DMM dataset | 1259 | 2144 | 4 | RS16E | point-wise |

### 2.4. Energy Splitting DMM Module

In the DMM module structure shown in Figure 6, the first step involves computing geometric operators for forestry feature point clouds. To accurately describe the characteristics of forestry point clouds, we utilize different point cloud feature descriptors compared to previous approaches. Since forestry point clouds do not include buildings or utility poles, we focus on linear feature descriptors, planar feature descriptors, and scattering feature descriptors, while excluding vertical feature descriptors. To characterize the point cloud, we employ a geometric approach for the point cloud descriptor. Equation (1) represents the linear geometric characteristics L of the point cloud, while Equation (2) measures P, the extent of linear stretching and elongation within the point cloud's neighborhood. Equation (3) evaluates S, the flatness of the point cloud, indicating its conformity to a plane, and considers scattering and the description of the divergence characteristics of point clouds. These three features collectively capture the dimensional properties of the point cloud.

$$\text{L} = \frac{\lambda_1 - \lambda_2}{\lambda_1} \tag{1}$$

$$\text{P} = \frac{\lambda_2 - \lambda_3}{\lambda_1} \tag{2}$$

$$\text{S} = \frac{\lambda_3}{\lambda_1} \tag{3}$$

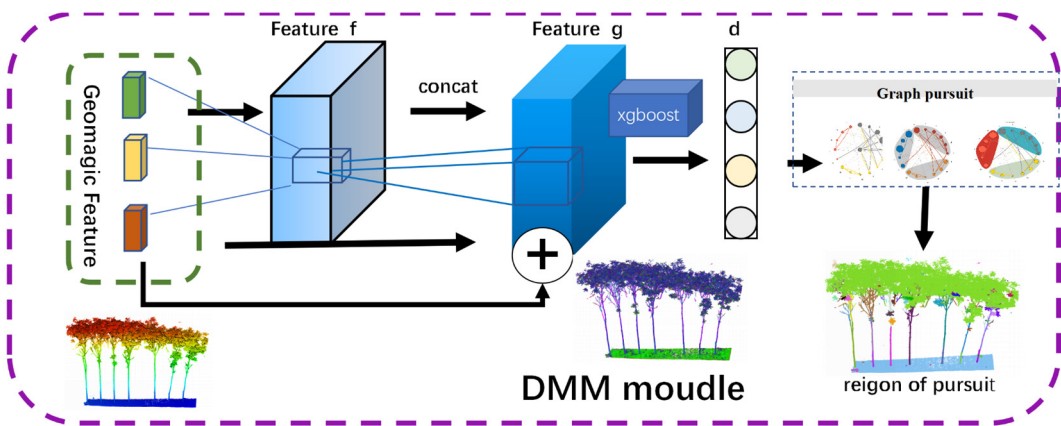

**Figure 6.** DMM module structure diagram.

We introduced the forestry feature descriptor 'R' to delineate tree characteristics [36]. For each point in the point cloud, calculate their linear characteristics $L_i$, plant characteristics $P_i$, and scattering characteristics $S_i$. As shown in the table below, calculate the forestry characteristic factor R and the terrain characteristic factor r for algorithm DMM. Trees exhibit linear aggregation in their branches and trunks, disperse needle-like leaves, and exhibit an overall vertical orientation relative to the coordinate plane. As a result, we defined the tree feature aggregation factor 'R' to encompass linearity, scattering, and elevation attributes within the point cloud. In contrast, ground point clouds in forestry environments often feature gullies, primarily characterized by planar traits and linearity. Conversely, shrubs, being closer to the ground, exhibit dispersive traits, and we aggregate scattering and planarity to describe them.

To automatically identify valuable information within the aggregated cascaded features of the DMM dataset, we applied the max pooling method using an attention mechanism. Additionally, we employed the XGBoost [37] feature filter to refine the selection of relevant features, resulting in the final point cloud feature description. The purple color in the representation corresponds to the tree feature factor 'R', while the green color signifies the terrain feature factor 'r'.

$$f_i = MLP(L_i, P_i, S_i) \tag{4}$$

$$g_i = concat(f_i, L_i, P_i, S_i) \tag{5}$$

$$r_i^k = MLP(p_i \oplus f_i \oplus g_i) \tag{6}$$

$$C_i^k k = b\left(r_i^k, w\right) \tag{7}$$

$$R_i^k = \sum_{k=1}^{i} \left(r_i^k \cdot C_i^k\right) \tag{8}$$

$$d\prime = \text{Shared} mlp(R_i^k) \tag{9}$$

$$L(\mu) = \underset{g \in R^d}{\text{argmin}} \sum \|N_i - n_i\|^2 + P\sum (N_i - N_j \neq 0) \tag{10}$$

As shown in the above formula, for each point in the point cloud $P = P_1 \cdots\cdots P_n$, calculate their linear characteristics $L_i$, plant characteristics $P_i$, scattering characteristics $S_i$, and connecting feature vectors. The new vector is obtained $f_i$, as well as connect features $g_i$ and given N points, and the KNN algorithm is used once for each point to find the nearest K Euclidean points The 3D coordinates of the central point, the 3D coordinates of the current point, the relative coordinates, and the European distance are connected. Then, the dimension is adjusted by MLP to adjust the vector of growth d to make the aggregation of the converged point cloud features $r_i^k$. Through the attention mechanism feature transformation, feature extraction, where W is the learnable weight of the shared MLP, $C_i^k k$ is obtained. Using the previously learned attention value to weight and sum, the attention value can be regarded as one that can automatically screen important information via soft mask. By filtering the surrounding point information, one can obtain the reduced feature vector $R_i^k$. After attention pooling, it becomes a vector of $d\prime$, and the goal of the CutPursuit algorithm is to minimize an energy function, which usually includes two main parts: a data fidelity term and a regularization $L(\mu)$. Complete each small block formed by over-segmentation, S = Cutpursuit Segmentation (L); finally, the point cloud is divided into several blocks. $S = S_1, \cdots S_j$. Each similar block has similar characteristics. "P" is part of the regularization term. This is a problem when finding the optimal energy.

Our dataset is divided into three partial point cloud feature extraction visualizations: DMM-1, DMM-2, and DMM-3, as shown in the figure above. The colors red, purple, and green represent linear, planar, and scattering features, respectively. In Figure 7, the top-left portion depicts a single tree point cloud feature visualization, where the branches reflect more linear characteristics (red), the canopy part exhibits scattering (purple), and the ground part appears planar (green). The top-right part of Figure 7 shows a small range visualization of several tree cloud features, where the branches predominantly exhibit linear characteristics (purple), the ground part demonstrates planarity (red), and the crown part shows scattering (green). The lower part of Figure 7 represents the visualization of cut point cloud features in large-scale forestry scenes, where the branches exhibit more linear characteristics (red), the crown part appears scattered (purple), and the ground part appears planar (green).

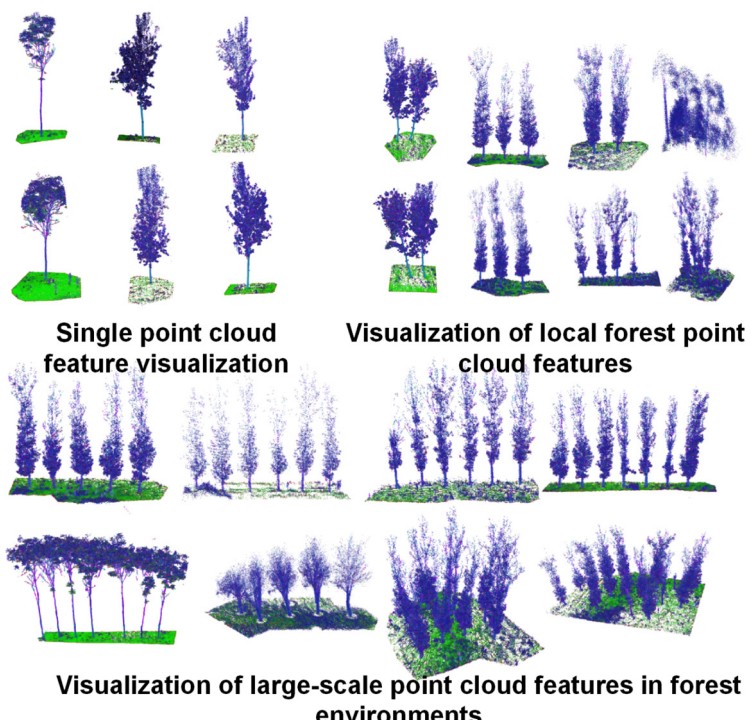

**Single point cloud feature visualization**

**Visualization of local forest point cloud features**

**Visualization of large-scale point cloud features in forest environments**

**Figure 7.** Visualization of DMM point cloud forestry feature extraction, where purple corresponds to the tree feature factor R and green represents the terrain feature factor r.

We present a comprehensive elucidation of the methodology that forms the foundation of the energy partitioning network. By utilizing computational techniques to process the raw input point cloud data, we transform a voluminous dataset consisting of millions of data points into a meticulously structured format characterized by hundreds of geometric partitions. Within each of these partitions, a remarkable degree of congruity can be observed in the local geometric attributes exhibited by the data points. The genesis of this intricate geometric partitioning scheme is firmly rooted in the 3D geometric features.

For the input original point cloud (denoted as P), the process of geometric partitioning is executed by leveraging its intrinsic 3D geometric attributes. The partitioning procedure considers the distinctive features inherent to each individual point within the point cloud. Notably, in this partitioning scheme, every data point is exclusively assigned to a singular geometric partition, precluding any overlap or multiple memberships. We employ the max pooling method to automatically learn the useful information in the aggregated cascaded features through the attention method and use the xGboost feature to filter the valid features. Finally, the new aggregated features are obtained. Combining the above features, we construct an unsupervised graph of the output features in the DMM module to construct an over-segmented unsupervised data point. For each individual data point, we utilize its local geometric feature vector as a representation that encapsulates the aggregated features discussed earlier. Our primary objective is to optimize the solution for L in order to achieve a resolution for the optimization problem as described in reference.

In our pursuit of solving this problem, we draw inspiration from the concept of greedy cutting, which is strategically applied to the 3D point cloud dataset. The following section outlines the energy-optimized procedural steps for aggregating the integrated 3D point cloud features.

Problem formulation: Our endeavor revolves around the minimization of the function L. The specific problem at hand necessitates the minimization of L&O [38], as proposed in a 2017 work introducing a working set strategy for minimizing differentiable functions constructed on weighted graphs, augmented with full variational half-table regularization.

We propose an enhanced algorithm that expands the algorithm's applicability to functions that contain non-differentiable segments distributed across the graph vertices, as illustrated on the left. In cases where function g demonstrates differentiability with respect to variable v, our algorithm identifies the locations of smooth points in function F. These smooth points are characterized by having zero differentials while possessing both positive and negative left and right derivatives. It is important to note that this assumption holds valid when all the considered general functions exhibit convexity, thereby making a smooth point equivalent to a global minimum value.

In the hyper-segmentation map of the forestry environment scene within the DMM dataset, it is evident that the trees collectively form a diverse array, while the terrain exhibits an excessive degree of segmentation, resulting in the fragmentation of the ground into numerous minutiae. This segmentation approach effectively discerns between towering trees and diminutive shrubbery. Notably, disparate objects are discretely partitioned into distinct segments, and sizable objects undergo further decomposition into smaller constituent elements. Figure 8, top left, depicts a single tree point cloud over segmentation visualization, revealing the division of a tree into different segments. In Figure 7, the top right part showcases a small range of trees over segmentation visualization. In this scenario, several trees with similar characteristics aggregate into the same category. Figure 8, lower part, presents a large-scale environment cut scene visualization, where the ground polymerization is improved, and many trees are synthesized into the same category.

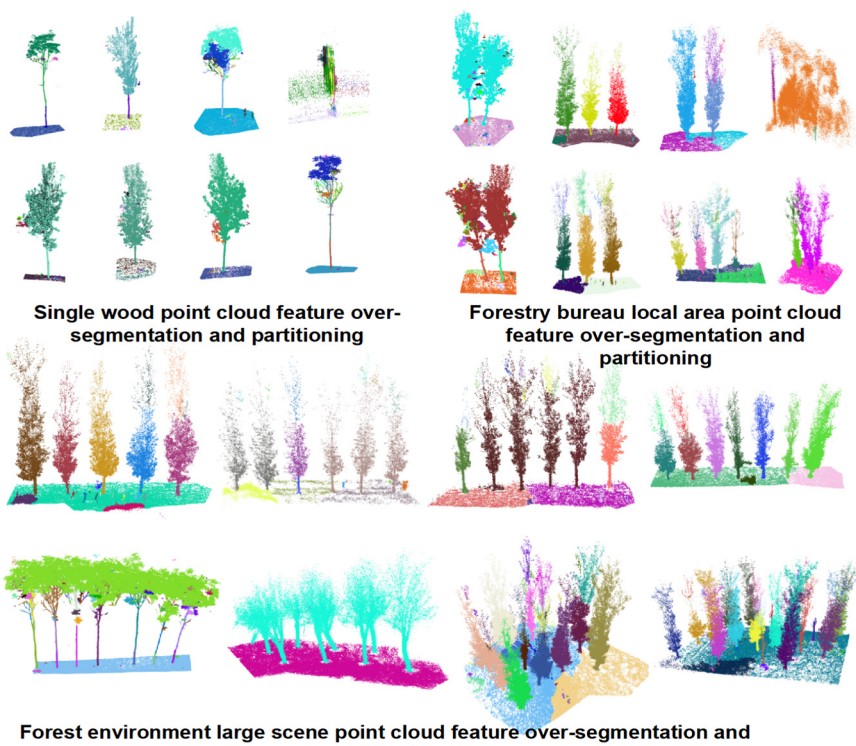

**Figure 8.** DMM dataset semantic hyper-segmentation.

## 2.5. PointDMM Net Network Structure

According to Figure 9, PoinDMM is an end-to-end network architecture. The key feature of the PoinDMM network architecture is its ability to directly process raw point cloud data and transform it into a feature representation with semantic information through a series of processing steps.

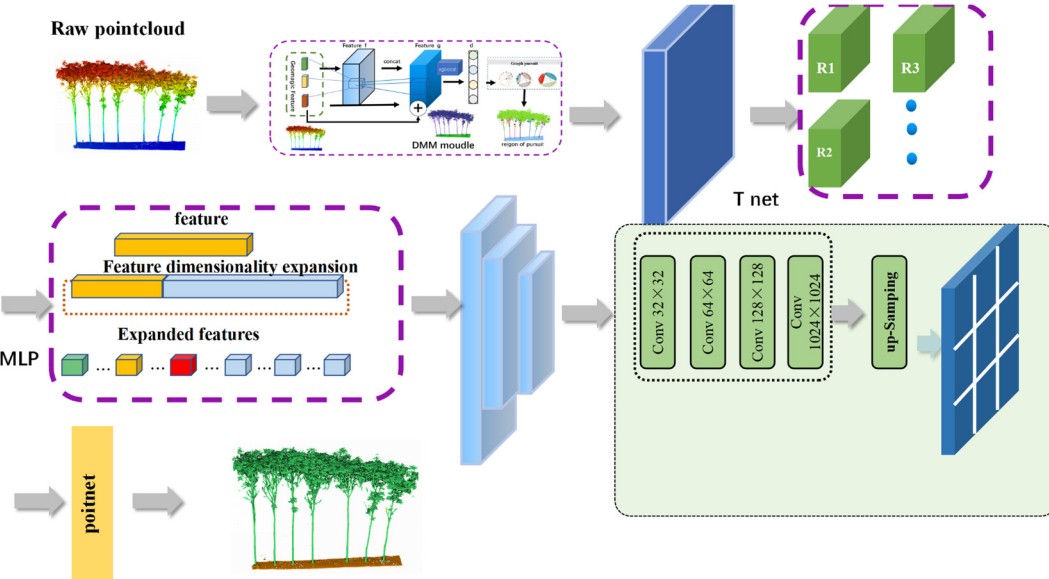

**Figure 9.** PointDMM deep learning network structure.

First, to meet the requirements of subsequent processing, the point cloud data need to be initialized and transformed into h5 format. This step converts the point cloud into batches of 8000 points for efficient batch processing operations, considering the limitation of computer memory.

Next, the point cloud data undergo processing through the DMM module mentioned in Section 2.4. The DMM module divides the point cloud into multiple regions based on semantic correlation and local forestry features. Features are then extracted for each region, allowing PoinDMM to capture detailed information in the point cloud data and improve its representation capability.

After feature extraction, PoinDMM employs the T network for rotation operations. This rotation operation, based on the method proposed by PointNet, improves the recognition accuracy of the point cloud by rotating it around the z-axis. By better processing the point cloud through rotation, PoinDMM enhances the accuracy of subsequent processing and analysis.

Following rotation, PoinDMM expands the dimension of point cloud features using a multi-layer perceptron (MLP). This step aims to avoid feature loss during computation. By expanding the feature dimension, PoinDMM retains more information, thereby enhancing the effectiveness of subsequent processing steps.

PoinDMM then performs convolution operations on the features. Through multiple layers of convolution operations, utilizing 32, 64, 128, and 1024-dimensional convolution layers, PoinDMM further extracts features from the point cloud data and expands them to different dimensions. This step increases the expressive power of the features and improves the capture of information in the point cloud data.

Finally, PoinDMM utilizes a sampling method to output the features. Through this sampling operation, PoinDMM reduces the dimension of the feature representation and performs semantic segmentation using PointNet, resulting in the final semantic segmentation result. The purpose of this step is to segment the point cloud data into different categories and assign corresponding semantic labels to each category.

In conclusion, PoinDMM is an effective network architecture for the analysis and processing of raw point cloud data. By directly processing the point cloud and extracting semantic information through a series of processing steps, PoinDMM produces a final semantic segmentation result. This makes PoinDMM a valuable tool for point cloud data analysis and processing.

## 3. Experiment and Result

### 3.1. Training Environment and Parameter Configuration

We delve into further intricacies of our training procedure. Detailed training parameters are presented in Table 3 below. It is important to note that all aspects of both training and testing were conducted on a personal computer, utilizing the computational power of an NVIDIA GeForce RTX 3060 graphics card GPU for CUDA-accelerated computation. Our development environment was meticulously configured, consisting of Python 3.6 and TensorFlow-GPU 2.4.1, operating seamlessly on the Ubuntu 18.04 platform.

**Table 3.** PointDMM parameter settings.

| Training Hyperparameters | Parameter Values |
|---|---|
| Initial KNN parameter of DMM module | 10;100;1000 |
| Maximum number of iterative steps | 500 |
| Study attrition rate | 0.7 |
| Base study rate | 0.02 |
| Batch study size | 3 |
| Learning momentum | 0.8 |
| Block Size | 150 |

We utilized an optimizer to automatically determine the optimal learning rate of 0.41. Subsequently, we employed a technique of gradually reducing the learning rate in small increments to identify the most favorable learning rate. Eventually, we established the final optimal learning rate as 0.45 and proceeded with pointDMM training. Throughout the training process, we observed an upward trend in training accuracy and a downward trend in the training loss function. These findings indicate that our network exhibits excellent learning capabilities for global features. The total duration of training and validation amounts to approximately 540 h. The DMM module and training are separate. Our program is divided into two parts. The DMM module generates segmented block features and block features, taking a total of 19,538 s. After that, the network training is performed. The output of the DMM module includes features, feature groups, original point cloud coordinates, and different blocks. After calculating and saving these results, our server can take a rest and enter the training phase. The network training takes 12,873 s, totaling 32,411 s, as shown in Table 4, After 500 training iterations, the accuracy and loss curves of the dataset tend to stabilize, with the training accuracy and loss function converging to 0.945 and 0.12, respectively.

**Table 4.** Computation time (seconds) of pointDMM for semantic segmentation on DMM dataset.

| Method | DMM Moudle Time | Train Time | Total Time |
|---|---|---|---|
| PointDMM | 19,538 | 12,873 | 32,411 |

### 3.2. DMM Datas

At the same time, the PointLAE algorithm was evaluated on the DMM dataset. A development environment was set up on Ubuntu 18.04, consisting of Python 3.7 and PyTorch 1.0. Point clouds possess the characteristic of rotational invariance, and to enhance the training process, data augmentation was performed by randomly rotating point clouds around the z-axis. Additionally, a random dropout technique was applied with dropout rates of 0.3, 0.5, and 0.7. This involved randomly removing some point clouds from the training set during each epoch. By incorporating random dropout, the generalization capability of the training process was effectively improved, resulting in excellent performance on sparse point clouds. The default parameters used in the experiments are presented in Table 5. As for other methods, I trained using the default configuration of the source code provided by the author on GitHub.

**Table 5.** PointLAE Testing on DMM dataset parameter settings.

| Training Hyperparameters | Parameter Values |
| --- | --- |
| Initial KNN parameter of DMM module | 10 |
| Maximum number of iterative steps | 500 |
| Study attrition rate | 0.65 |
| Base study rate | 0.002 |
| Batch study size | 2 |
| Learning momentum | 0.9 |
| Block Size | 50 |

### 3.3. Evaluation Indicators

According to the semantic3d evaluation metrics we adhere to, we employed several key metrics to evaluate the dataset used. These metrics encompass the application recall IoU, IoU per class, joint intersection, and overall accuracy OA. The application recall IoU assesses the number of samples predicted as class j from the class i group structure. The IoU per class evaluates the accuracy of each individual class. The joint intersection metric is employed to evaluate the intersection between all classes. The overall accuracy OA is utilized to gauge the overall accuracy of the dataset. C represents the number of samples predicted as class j from the class i group structure. These evaluation metrics are pivotal for assessing the performance of the dataset. By employing the recall IoU metric, we can comprehend the prediction accuracy of the model across different classes. The IoU metric per class aids in understanding the prediction accuracy of each class, thereby facilitating a better grasp of the model's performance across different classes. The joint intersection metric furnishes information about the intersection between different classes, enabling us to comprehend the relationship between classes. The overall accuracy OA metric is a comprehensive evaluation metric that assists in comprehending the overall prediction accuracy of the entire dataset. Through a comprehensive analysis of these evaluation metrics, we can more comprehensively evaluate the performance of the dataset used and further optimize the model's performance. These evaluation metrics provide valuable information that helps us comprehend the model's performance across different classes and provide directions for improving the model. These evaluation metrics are indispensable tools for dataset evaluation and model optimization.

$$IoU_i = \frac{c_i}{c_{ii} + \sum\limits_{j \neq i} c_{ij} + \sum\limits_{k \neq i} c_{ki}}. \tag{11}$$

$$A\_IoU = \frac{\sum\limits_{i=1}^{L} IoU_i}{L} \tag{12}$$

$$OA = \frac{\sum\limits_{i=1}^{L} c_{ii}}{\sum\limits_{i=1}^{L} \sum\limits_{k=1}^{L} c_{jk}} \tag{13}$$

### 3.4. Experimental Results

In Figure 10, the accuracy and level of detail in the predicted results are evident. Each pixel is accurately segmented into its corresponding category, enabling precise understanding and analysis of the image. By dividing the training and validation sets, we can verify the stability and reliability of our algorithm on different datasets.

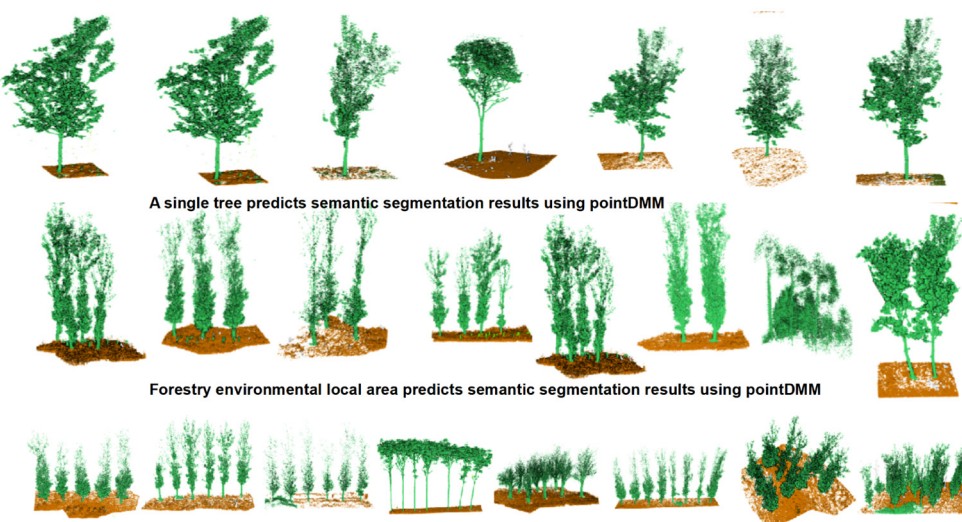

**Figure 10.** Visualization of the segmentation effect of pointDMM on the DMM dataset.

On the single-tree DMM-1 dataset, our pointDMM algorithm accurately identifies and segments individual tree structures. This achievement is of great significance for the management and protection of forest environments as it helps us gain a better understanding of tree growth and health.

On the local forestry environment area DMM-2 dataset, our algorithm effectively identifies different vegetation types and terrain features. This capability is crucial for the management and planning of forest resources, as it helps us better understand the structure and composition of forests and enables us to take appropriate protection and management measures.

On the large-scale forest environment point cloud DMM-3 dataset, our algorithm can handle a large amount of point cloud data and accurately segment different objects and landscape features. This capability is vital for monitoring and evaluating forest resources, as it helps us better understand the overall condition of forests and the health of ecosystems.

Overall, our pointDMM algorithm performs well on different datasets, demonstrating high accuracy and stability. These results provide strong support for further research and the application of semantic segmentation technology. Additionally, they serve as an important reference for forest resource management and environmental protection. Through these research achievements, we can gain a better understanding of the natural environment and promote sustainable development.

As depicted in Figure 11c, the overall accuracy of DMM-3 is 0.93 in the three test datasets, which is significantly higher than Figure 11a, which shows the accuracy of DMM-1 (0.88), and DMM-2 (0.85), as shown in Figure 11b. This improvement may be attributed to the larger amount of data used in DMM-3 and its ability to better capture the point cloud features of multi-scale forest environments through pointDMM learning. Compared to the other three algorithms, DMM-3 has shown a 21% increase in accuracy. Among the three datasets, the algorithm performs best in classifying trees, followed by the ground, indicating its effectiveness in extracting point cloud features of forest stands. However, the accuracy is lower for miscellaneous point clouds, which mainly consist of ground stones, fallen logs, and other materials. These points are fewer in number and the network struggles to learn their features, resulting in misclassification.

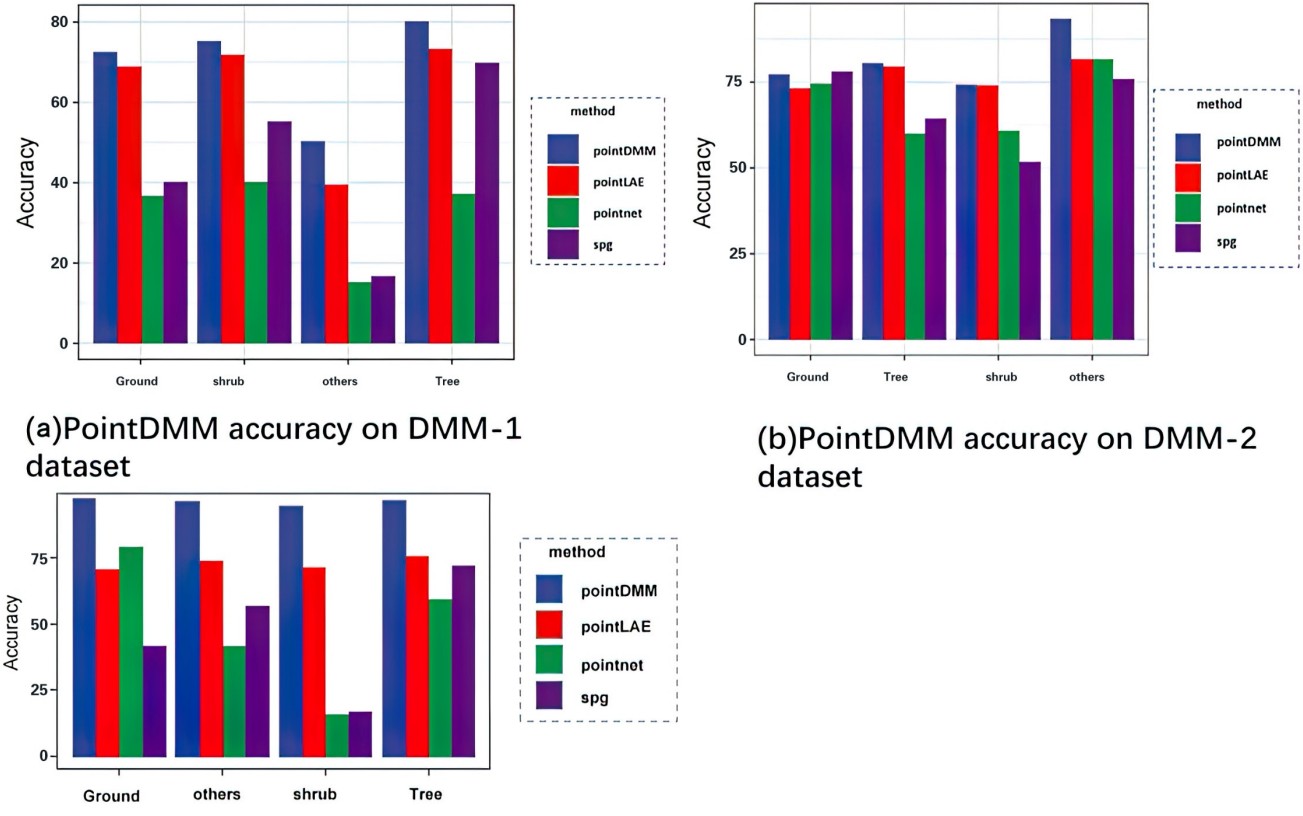

(a)PointDMM accuracy on DMM-1 dataset

(b)PointDMM accuracy on DMM-2 dataset

(c)PointDMM accuracy on DMM-3 dataset

**Figure 11.** The accuracy of pointDMM on the DMM dataset.

## 4. Discussion

### 4.1. Method Evaluation

Forest resource surveys traditionally rely on plot survey methods, which are known for being time-consuming, labor-intensive, and prone to low accuracy. However, the utilization of laser scanning technology for forestry scene analysis has brought about significant advancements in forestry automation, intelligence, and digitization. In this study, a precise annotation DMM dataset is constructed for the purpose of multi-forest identification and segmentation, utilizing a forestry environment mobile information collection platform. This dataset has proven to be highly efficient for handling large-scale and multi-data volume forestry scenes, effectively addressing challenges posed by point cloud occlusion and uneven sparsity within forestry environments. Additionally, the study proposes the implementation of the DMM module, which serves as a means of semantic feature pre-segmentation for forest scenes. Furthermore, an end-to-end deep learning method called pointDMM is introduced for point cloud semantic segmentation, achieving impressive segmentation results with a high level of robustness. However, it is important to acknowledge certain limitations within this work. Due to the complexity of undergrowth resources in forests, such as fallen trees and debris, manual labeling of point clouds remains highly subjective. While the labeling of the ground and trees is typically accurate, errors may occur when labeling debris and low shrubs. Moreover, the choice of K-nearest neighbor initialization during training within the DMM module can impact both accuracy and system processing time. Overall, the forest information extraction method explored in this paper effectively captures the features of living trees, while the forest point cloud recognition method demonstrates strong performance in final testing. Through iterative training, the network obtains optimal weights, resulting in a robust model for point cloud recognition.

### 4.2. Parameter Selection of PointDMM

As depicted in Table 3, our pointDMM model incorporates three default parameters: k = 10, k = 100, and k = 1000. The processing speed of the point cloud increases as the value of k decreases. The selection of this parameter depends on the scale of the environmental scene. It is preferable to choose appropriate parameters to train the network for different environmental scenes. We conducted experiments on various environmental scene datasets, including the DMM-1, DMM-2, and DMM-3 datasets. The corresponding results are visualized in Figures 12–14.

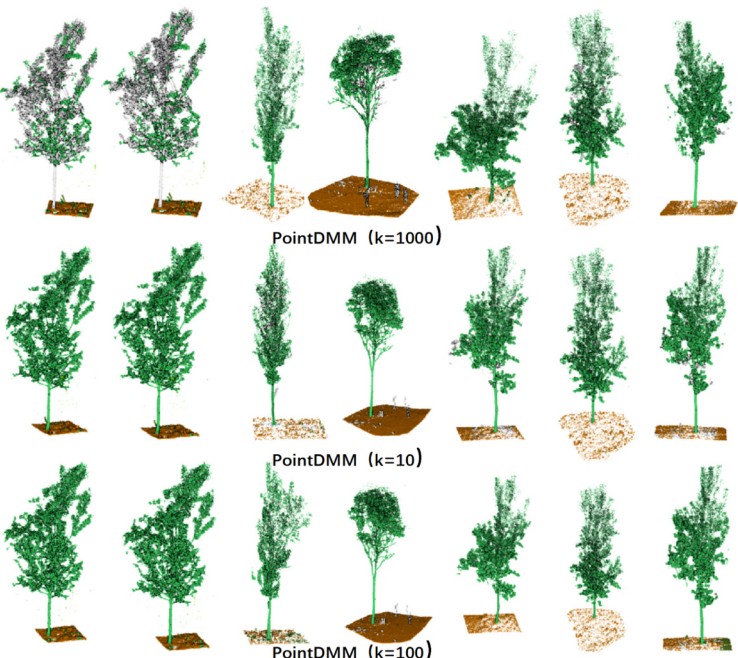

**Figure 12.** Visualization of the semantic segmentation of different parameters of pointDMM on the DMM-1 dataset.

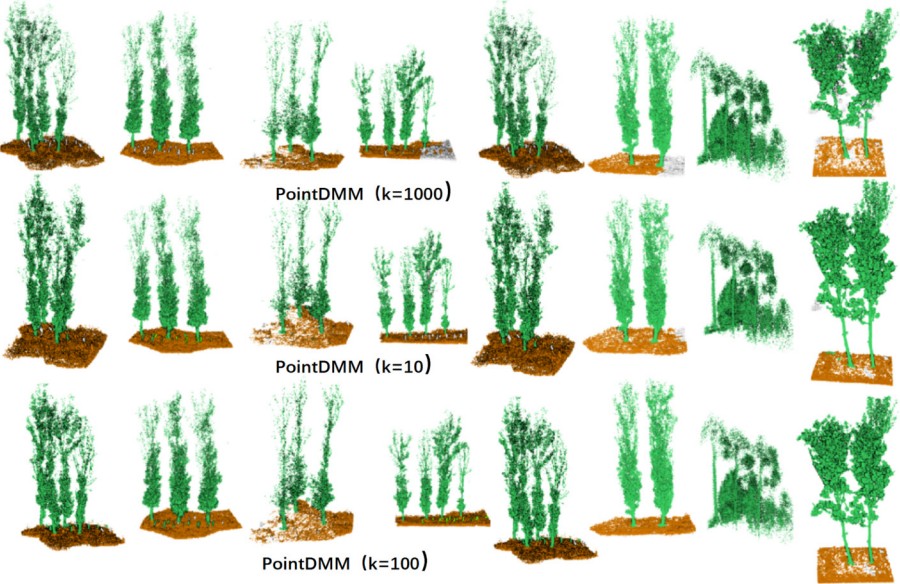

**Figure 13.** Visualization of the semantic segmentation of different parameters of pointDMM on the DMM-2 dataset.

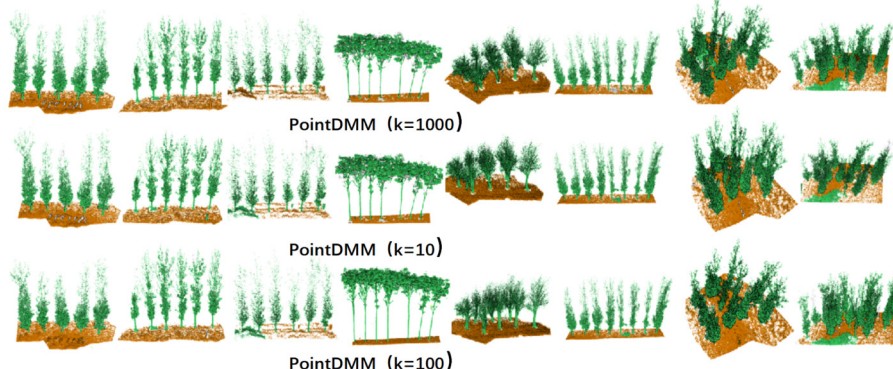

**Figure 14.** Visualization of the semantic segmentation of different parameters of pointDMM on the DMM-3 dataset.

For the DMM-1 dataset, as shown in Figure 12, the point cloud classification performance for individual trees is not satisfactory when selecting k = 1000. This could be attributed to the large scale of the dataset, which makes it challenging to capture the local geometric features of single-tree scenes. As observed in the figure, the point cloud is segmented into various classes. However, in the DMM-2 and DMM-3 datasets, the point cloud performance is significantly improved. In the DMM-1 dataset, the highest classification accuracy is achieved at k = 10, but it comes at the cost of a long processing time of 622 h. On the other hand, selecting k = 100 reduces the processing time to 355 h, with a slight decrease in accuracy of 1.2%. In Figures 13 and 14, the three accuracy performances are comparable. However, compared to selecting k = 10, choosing k = 100 improves efficiency by 33%, and selecting k = 1000 improves efficiency by 48%.

### 4.3. Compare PointDMM with Other Algorithms

We combine the DMM-1 dataset, the DMM-2 dataset, and the DMM-3 dataset to create the DMM reduced dataset and train it. As depicted in Figure 15, our pointDMM algorithm demonstrates excellent performance. The overall accuracy reaches 83.25%, showcasing significant advancements in the recognition of standing trees compared to other algorithms. Specifically, there is an 11% improvement compared to pointLAE [39], as well as a 2% increase in accuracy for shrub identification. Considering performance, the pointDMM algorithm with k = 10 is recommended for small-scale forestry scenes, such as individual tree surveys. For medium-scale plot surveys, the pointDMM algorithm with k = 100 is preferred, while for large-scale forest environment surveys, the pointDMM algorithm with k = 1000 is utilized.

In the future work of this project, our proposed point cloud recognition network relies heavily on a large number of GPUs for computation. However, in intelligent forestry applications, only limited information, such as pre-planning, can be provided. Therefore, it is meaningful to explore the deployment of lightweight hosts such as NVIDIA for real-time forestry point cloud recognition in mobile measurement and collection scenarios. By using lightweight hosts, we can reduce costs, improve efficiency, and better adapt to the needs of mobile measurement and collection in intelligent forestry. In this paper, our main focus is on the processing of ground laser point clouds. With the development of ALS technology, the use of drones for large-scale data collection and the fusion of drone data with ground point clouds can enhance the algorithms for better handling of canopy features in airborne LiDAR point clouds. This improvement will make our algorithms more suitable for airborne LiDAR point cloud collection systems, which is an important area for further exploration. In future work, we can further study how to optimize the processing speed of the point cloud recognition network and reduce its dependence on GPUs. Parallel computing techniques, such as distributed computing, can be considered to improve processing speed. Additionally, we can explore how to further optimize the performance of lightweight hosts to meet the real-time point cloud recognition needs in

mobile measurement and collection. Furthermore, we can conduct further research on methods for data fusion of drone data collection and ground point clouds. This research can focus on effectively fusing point clouds collected by drones with ground point clouds to improve the accuracy and robustness of recognition algorithms. Moreover, we can explore how to more accurately recognize canopy features in airborne LiDAR point clouds to further enhance algorithm performance. This research can contribute to better understanding and the utilization of airborne LiDAR data in intelligent forestry applications. In summary, future work can focus on optimizing the processing speed of the point cloud recognition network, improving the performance of lightweight hosts, researching methods for data fusion of drone data collection and ground point clouds, and more accurately recognizing canopy features in airborne LiDAR point clouds. These efforts will further promote the application of intelligent forestry in mobile measurement and collection, providing more accurate and efficient auxiliary means for forestry management and protection.

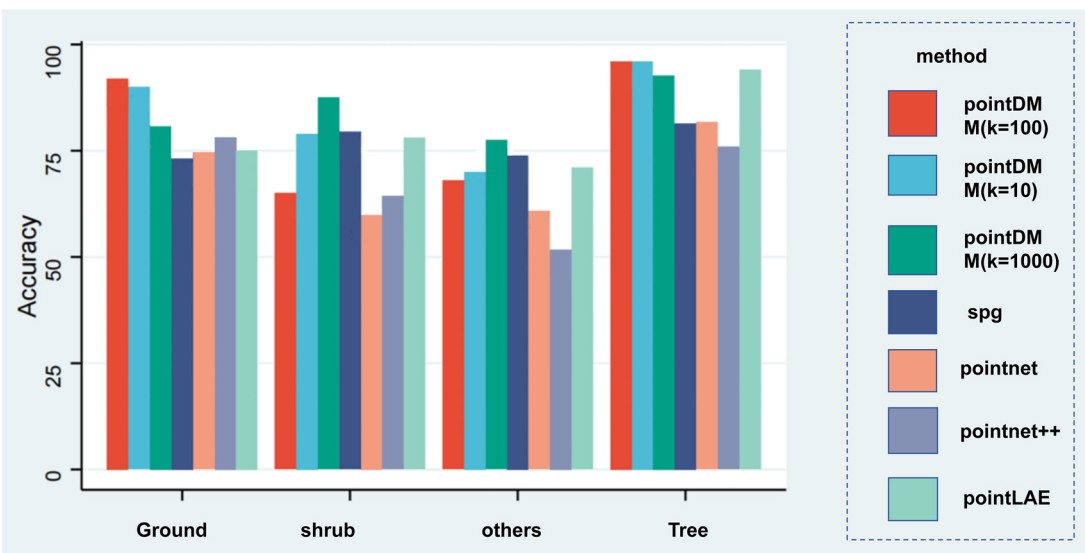

**Figure 15.** Accuracy of the semantic segmentation of different parameters PointDMM and other method on the DMM reduced dataset.

## 5. Conclusions

This work proposes a ground-based LiDAR point cloud semantic segmentation method for complex forest undergrowth environments. Our work consists of three main components: the construction of forestry point cloud datasets, the fusion of undergrowth point cloud features, and semantic segmentation using the DMM module based on a deep learning method called pointDMM. The forestry dataset is collected using backpack-style LiDAR devices. To address the lack of a database, large scene scales, and high data volume in forestry resource environment point clouds, we propose a point cloud data annotation method based on single-tree positioning to improve annotation efficiency and address difficulties caused by occlusion and sparse distribution in forestry environments. To address the characteristics of diversity, disorder, large data volume, large scene scales, and uneven sparsity in forestry environments, as well as the less ideal fractal structures, we integrate tree features using the DMM module and construct a critical segmentation graph using an energy segmentation function. We then use cutpursuit to solve the graph and achieve pre-segmentation of semantics. Our method fills the gap in existing deep models for complex forestry environment point cloud information, which is characterized by severe occlusion, high density, complex terrain, multiple return information, and uneven scales. We propose an end-to-end deep learning model called pointDMM, which trains a multi-level lightweight deep learning network to effectively improve the intelligent analysis of complex forestry environment scenes. Our method demonstrates good segmentation

results on the DMM dataset, achieving an accuracy of 93% on the large-scale forest environment point cloud dataset DMM-3, which has a 21% improvement in live tree identification accuracy compared to other algorithms. Compared to the manual segmentation of point clouds, this method has significant advantages in extracting feature information from TLS-acquired artificial forest point clouds, laying a solid foundation for achieving automation, intelligence, and informatization in forestry.

**Author Contributions:** Conceptualization, J.L. (Jinhao Liu); Methodology, J.L. (Jiang Li); Software, Q.H.; Formal analysis, J.L. (Jinhao Liu); Writing—original draft, J.L. (Jiang Lia); Project administration, J.L. (Jinhao Liu); Funding acquisition, J.L. (Jinhao Liu). All authors have read and agreed to the published version of the manuscript.

**Funding:** This research is supported by Natural Science Foundation of China (No. 62006008, 62173007, 61903009), and National Key Technology R&D Program of China (No. 2021YFD2100605).

**Data Availability Statement:** The data used to support the findings of this study are available from the corresponding author upon request.

**Conflicts of Interest:** The authors declare no conflict of interest.

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
