# Peer review of "PointDMM: A Deep-Learning-Based Semantic Segmentation Method for Point Clouds in Complex Forest Environments"

_forests, doi:10.3390/f14122276_

Round 1

Reviewer 1 Report (Previous Reviewer 2)

Comments and Suggestions for Authors

A pipeline for semantic segmentation of PCs is described. The methodology gives rather good performance values on chosen sites and data sets, but it is hard to estimate how well this carries to different environments with less organization in the forest structure. 

THe physical site used seems to be such where a backpack campaign is reasonable, since the tree density does not allow vehicle based scanning. Although, an UAV scanning could be more economical (but more difficult to achieve results). 

Comments:

ground-based radar point cloud --> ground-based laser scanner point cloud.

Change "radar" to LiDAR (light detection and ranging) or "laser scan" everywhere.

p. 11, 2. Denoising: You should either use boldface for this "subsubsection" name, or make it easy to find (check instructions for aothors) Same with previous topic 1.: "Semantic-Segmentation Editing"

P. 11-12: Reveal the algorithmic structure better in topics 2-5. You don't need to  go to the complete format of DMM module algorithm (p. 15), but present steps in separate lines and make mathematical formula readable. 

p. 13: Table 2: make this one narrow and clear by e.g. giving sensors shorthand symbols in the text and the caption text: V64: Velodyne 64 ...,

: SIC, SICK ... and so on. Change "Points (Millions)" to "Points \\ ($10^6$)

P. 14: Moudule --> Module

Introduce L, P and S before the equations, Make sure all variables, 'R', and ones in the Topics 2-5 algorithm descriptions really are types as mathematical variables. Use proper names fro L,P,S everywhere: 
(linearity, planarity, sphericity, see https://www.researchgate.net/publication/311630470_A_Bayesian-Network-Based_Classification_Method_Integrating_Airborne_LiDAR_Data_With_Optical_Images)

p. 16: I think usage of P(.): shoukd be like this: 
argmin ... + \lambda \sum_{(i,j)\in ??? } P(N_i \neq N_j) ) 

where P(false)= 0 and P(true) = 1 (introducing a predicate function P)

and \lambda is the regularization weight (there has to be this one?). But if I have understood it wrong, then most of the readers woukd do the same, and then it is better for you to explain it better.

4 Discuss --> Discussion

4.1 Evaluate our method --> Method evaluation

5 conclusion --> Conclusion

Author Response

Responds to the reviewers’ comments

Dear Editors and Reviewers:

Thank you for your letter and for the reviewers’ comments concerning our manuscript entitled (forests-2690468). Those comments are all valuable and very helpful for revising and improving our paper, as well as the important guiding significance to our researches. We have studied comments carefully and have made a correction which we hope meets with approval. We tried our best to improve the manuscript and made some changes to the manuscript according to your suggestions. These changes are marked in highlighted style, which will not influence the content and framework of the manuscript.

We appreciate for editors' and reviewers’ warm work earnestly and hope that the correction will meet with approval. The main corrections in the paper and the responses to the reviewers’ comments are as following:

Reviewer #2:

General comments:

Dear Authors,

  The manuscript "PointDMM:A deep learning-based semantic segmentation method for point clouds in complex forest environments" presents an interesting topic that deals with the integration of data derived from Lidar scanning and that still has interesting challenges to overcome. In this context, you present an innovative methodology for processing a Lidar cloud with billions of points using deep learning and semantic segmentation.

   Considering the first version, you presented in this second version information from data collection, processing and visualization of the results.

     However, this ended up translating into a long manuscript with a very dense reading which can confuse the reader in some passages. I have tried to highlight these passages and you can consult them in the comments of the digital file.

       In chapter 1 you presented a good literature review on Lidar technology, however, the ideas are not well connected and need to be revised. The correct term is Laser/Lidar and not Radar. Please revise.

     A total of forty-eight bibliographical references are presented, which could be better explored in the presentation of results and discussions.

        The methodology was very detailed and extensive. I believe that a revision could reduce it without discarding important information.

The results are presented adequately, but the discussions could be improved by comparing the proposed algorithm with those developed by other authors.

Response:We are very grateful for your positive comments on the manuscript. We are honored that our approach has aroused your interest. Your positive comments are the greatest encouragement to us. Thank you for pointing out some errors in our paper.  At the same time, thank you for your valuable suggestions. I have replaced "radar" with "lidar" throughout the entire text, highlighted the references that can be deleted, and provided a better discussion of the results.The detailed corrections can be found in the detailed comments below.

    Comments:

  • Abstract: revise because the problem is not correctly presented. Highlight the article's contribution to solving the problem. 

Response:Thank you very much for your suggestion. I have rewritten the abstract section, highlighting it from line 9 to line 41 on the first page.

  • Line 150: "meticulous calibration processes and manual data fusion in subsequent stages, thereby complicating the data processing pipeline". This statement is not correct because current Lidar systems have technology that processes the Lidar point cloud with the optical/image data automatically. 

Response:Thank you very much for your valuable suggestions. I have rewritten the first section, and the introduction in this part has been removed in the new version as it did not directly address forestry LiDAR point clouds.

  • The wording of chapter 1 should be revised. In some parts of the text, the ideas presented are not connected. 

Response:Thank you very much for your suggestion. I have made modifications and simplifications to the first section. My approach is to first explain the role and significance of point clouds in forestry (lines 46 to 77 on page 2), then introduce the current research status of point clouds in forestry, and further elaborate on the difficulties of traditional methods for processing point clouds, leading to the development of five directions of deep learning in point cloud processing (lines 159 to 206 on page 4). Finally, I discuss our research by highlighting the problems encountered in the application of point cloud deep learning methods in forestry (lines 207 to 224 on page 5). At the same time, irrelevant literature introductions were deleted from this research method.These modifications are marked in the document.

  • Standardize the units presented. For example, mu and acres are used for area values. Hectares is a more common unit.

Response:Thank you for your valuable suggestions. In the document, I have already replaced acres with hectares and highlighted it on page 6, lines 246 and 266.

  • Line 306: "Baidu cloud link" please provide the download link used.

Response:Thank you for your suggestion. Due to the large amount of data, I have organized all the experimental results mentioned in this document. The point cloud of the intermediate steps in the experiment has a total size of 8.8GB. I have uploaded it to Quark Cloud Drive, please see: https://pan.quark.cn/s/fdce3d6aedac. The relevant information can be found on pages 6, lines 266 to 270, and is highlighted in the main text.

  • When acronyms are presented first in the text, please give their meaning.

Response:Thank you again for your valuable suggestion. I have reviewed the entire text and defined the meanings of the abbreviations before them.

  • Table 1: please provide the limit line at the bottom of the table.

Response:Thank you for your valuable suggestion. Table 1 contains the hardware parameters of the laser radar equipment used for data collection. I have made the necessary modifications to make it more standardized. The updated table can be found on page 8, line 325, and is highlighted.

  • Line 399: is this the application used? https://github.com/GerasymenkoS/semantic-segmentation-editor

Response:Special thanks for your suggestion. This is the program we use as a benchmark for labeling categories. And it is explained on page 11, line 384 of the article, with highlighted markings.

  • Line 464: "Take a small range (e.g. 5 centimeters) above and below the breast height position." Is this range adequate? This value is defined according to the density of the point cloud.

Response:Thank you for your valuable suggestion. The suggested modifications have been made in the document, and a highlight has been added on page 12, line 448.

  • Line 473: "5.CHM is short for Canopy Height Model". Normally, Lidar operated close to the ground has an inherent difficulty in capturing points above leaves and branches.

Response:Thank you for your suggestion. Indeed, it is difficult for ground-based LiDAR to capture canopy information. However, our data is obtained through LiDAR SLAM, which includes most of the tree canopy information. Please refer to the original point cloud in Figure 3 for more details.

  • Line 481: "Remove noise using point cloud denoising algorithms. Ground point extraction: Extract ground points from the point cloud using ground segmentation algorithms, such as progressive least squares or other methods.". What algorithms and parameters are used to remove the noise and classify the points on the ground?

Response:Thank you very much for your valuable suggestions. I apologize for the repetition. Denoising and ground point normalization have already been explained from page 11, line 386 to line 428. I have already deleted the relevant parts.

  • Line 494: in this item you present information that will be detailed in chapter 2.4.

Response:Thank you very much for your valuable suggestions. This is caused by repetitive introductions. I will rephrase it and refer to pages 12, lines 480 to 489, and highlight them.

  • Table 2/3: please check for extraneous fonts in the body of the tables.

Response:Thank you very much for your valuable suggestion. I have made modifications to Table 2, as seen on page 13, row 495, and highlighted it.

  • Line 555: "describing isotropic characteristics". Please detail which characteristics are derived.

Response:Thank you very much for your valuable suggestion. I apologize for not describing it clearly before. This refers to the description of point cloud divergent features. For more details, please refer to reference 47.

Guinard, S., & Landrieu, . “Weakly supervised segmentation-aided classification of urban scenes from 3D LiDAR point clouds.  ISPRS Workshop ,vol.3,pp.1321,2018

I provided a detailed description on lines 534 to 539 of page 14, and highlighted this section.

  • Figure 7: move it after the table and before the explanation (line 587)

Response:Thank you again for your valuable suggestion. I have moved Figure 7 to page 15 and highlighted it after paragraph 595.

  • Line 718: "The total training and validation time is approximately 540 hours". Did the processing take place uninterruptedly for 22.5 days?

Response:Thank you very much for your valuable suggestions. Our program is divided into two parts. The DMM module generates segmented block features and block features, taking a total of 19538 seconds. After that, the network training is performed. The output of the DMM module includes features, feature groups, original point cloud coordinates, and different blocks. After calculating and saving these results, our server can take a rest and enter the training phase. The network training takes 12873 seconds, totaling 32411 seconds, as shown in Table 4.I made table 4,Computation time (seconds) of pointDMM for semantic segmentation on DMM dataset,Specific description can be found on pages 20, lines 729 to 735. Table 4 is on page 21, line 739. This section is highlighted.

  • Figure 11: please configure the graphs in the same way. The third graph displays the values in bars. 

Response:Thank you for your valuable suggestion. I have made the modifications to the new figure 11, highlighting it on page 24, line 845.

  • It would be interesting if you could use letters in the figures to explain them more efficiently. For example, in Figure 11a you can say that...

Response:Thank you again for your valuable suggestion.I have made the modifications to the new figure 11, highlighting it on page 24, line 845.Describe simultaneously on pages 845 to 848 and highlight.

  • You have tested other algorithms to compare them with the DMM and their acronyms are shown in the figure legends. Please cite the algorithms tested and their configurations. This could be added to chapter 3.1

Response:The PointLAE algorithm was evaluated on the  DMM dataset. A development environment was set up on Ubuntu 18.04, consisting of Python 3.7 and PyTorch 1.0. Point clouds possess the characteristic of rotational invariance, and to enhance the training process, data augmentation was performed by randomly rotating point clouds around the z-axis. Additionally, a random dropout technique was applied with dropout rates of 0.3, 0.5, and 0.7. This involved randomly removing some point clouds from the training set during each epoch. By incorporating random dropout, the generalization capability of the training process was effectively improved, resulting in excellent performance on sparse point clouds. The default parameters used in the experiments are presented in Table 4.As for other methods, I trained using the default configuration of the source code provided by the author on GitHub. I made a table 4 describing the detailed configuration of pointLAE, with specific content on pages 21 from line 742 to line 770, highlighted.

  • Chapter 4: it would be interesting to comment on the results obtained with the other algorithms and the superiority achieved by using the DMM.

Response:Thank you for your meaningful suggestions. As a result, we have added Chapter 4.3 to evaluate and compare it with other methods on the DMM dataset. The specific content can be found on pages 26, lines 910 to 940.

  • The conclusions begin on line 907. The information presented in lines 894 to 906 ends up being repetitive.

Response:Thank you for your meaningful suggestions. I have made revisions to the conclusion section by removing redundant content and improving the wording. Please refer to the highlighted paragraph starting from line 954 on page 28 in the manuscript.

    I end my review by congratulating you on your hard work.

Respectfully,

  The wording needs a thorough review of punctuation, especially commas. Passages where the wording should be improved have been highlighted in the comments on the digital file. Be careful with the use of unusual terms in the English language. I recommend that the final version of the manuscript be proofread by a native English speaker.

Response:Thank you very much for your valuable suggestions and hard work in reviewing. I apologize for not being careful enough in my previous work. I have reviewed the punctuation and made overall language improvements. Thank you.

Reviewer 2 Report (Previous Reviewer 1)

Comments and Suggestions for Authors

Dear Authors,

  The manuscript "PointDMM:A deep learning-based semantic segmentation method for point clouds in complex forest environments" presents an interesting topic that deals with the integration of data derived from Lidar scanning and that still has interesting challenges to overcome. In this context, you present an innovative methodology for processing a Lidar cloud with billions of points using deep learning and semantic segmentation.

   Considering the first version, you presented in this second version information from data collection, processing and visualization of the results.

     However, this ended up translating into a long manuscript with a very dense reading which can confuse the reader in some passages. I have tried to highlight these passages and you can consult them in the comments of the digital file.

       In chapter 1 you presented a good literature review on Lidar technology, however, the ideas are not well connected and need to be revised. The correct term is Laser/Lidar and not Radar. Please revise.

     A total of forty-eight bibliographical references are presented, which could be better explored in the presentation of results and discussions.

        The methodology was very detailed and extensive. I believe that a revision could reduce it without discarding important information.

    The results are presented adequately, but the discussions could be improved by comparing the proposed algorithm with those developed by other authors.

       Below are my main suggestions: 

1) Abstract: revise because the problem is not correctly presented. Highlight the article's contribution to solving the problem. 

2) Line 150: "meticulous calibration processes and manual data fusion in subsequent stages, thereby complicating the data processing pipeline". This statement is not correct because current Lidar systems have technology that processes the Lidar point cloud with the optical/image data automatically. 

3) The wording of chapter 1 should be revised. In some parts of the text, the ideas presented are not connected. 

4) Standardize the units presented. For example, mu and acres are used for area values. Hectares is a more common unit.

5) Line 306: "Baidu cloud link" please provide the download link used.

6) When acronyms are presented first in the text, please give their meaning.

7) Table 1: please provide the limit line at the bottom of the table.

8) Line 399: is this the application used? https://github.com/GerasymenkoS/semantic-segmentation-editor

9) Line 464: "Take a small range (e.g. 5 centimeters) above and below the breast height position." Is this range adequate? This value is defined according to the density of the point cloud.

10) Line 473: "5.CHM is short for Canopy Height Model". Normally, Lidar operated close to the ground has an inherent difficulty in capturing points above leaves and branches.

11) Line 481: "Remove noise using point cloud denoising algorithms. Ground point extraction: Extract ground points from the point cloud using ground segmentation algorithms, such as progressive least squares or other methods.". What algorithms and parameters are used to remove the noise and classify the points on the ground?

12) Line 494: in this item you present information that will be detailed in chapter 2.4.

13) Table 2/3: please check for extraneous fonts in the body of the tables.

14) Line 555: "describing isotropic characteristics". Please detail which characteristics are derived.

15) Figure 7: move it after the table and before the explanation (line 587)

16) Line 718: "The total training and validation time is approximately 540 hours". Did the processing take place uninterruptedly for 22.5 days?

17) Figure 11: please configure the graphs in the same way. The third graph displays the values in bars. 

18) It would be interesting if you could use letters in the figures to explain them more efficiently. For example, in Figure 11a you can say that...

19) You have tested other algorithms to compare them with the DMM and their acronyms are shown in the figure legends. Please cite the algorithms tested and their configurations. This could be added to chapter 3.1

20) Chapter 4: it would be interesting to comment on the results obtained with the other algorithms and the superiority achieved by using the DMM.

21) The conclusions begin on line 907. The information presented in lines 894 to 906 ends up being repetitive.

    I end my review by congratulating you on your hard work.

Respectfully,

Comments on the Quality of English Language

    The wording needs a thorough review of punctuation, especially commas. Passages where the wording should be improved have been highlighted in the comments on the digital file. Be careful with the use of unusual terms in the English language. I recommend that the final version of the manuscript be proofread by a native English speaker.

Author Response

Responds to the reviewers’ comments

Dear Editors and Reviewers:

Thank you for your letter and for the reviewers’ comments concerning our manuscript entitled (forests-2690468). Those comments are all valuable and very helpful for revising and improving our paper, as well as the important guiding significance to our researches. We have studied comments carefully and have made a correction which we hope meets with approval. We tried our best to improve the manuscript and made some changes to the manuscript according to your suggestions. These changes are marked in highlighted style, which will not influence the content and framework of the manuscript.

We appreciate for editors' and reviewers’ warm work earnestly and hope that the correction will meet with approval. The main corrections in the paper and the responses to the reviewers’ comments are as following:

Reviewer #1:

General comments:

A pipeline for semantic segmentation of PCs is described. The methodology gives rather good performance values on chosen sites and data sets, but it is hard to estimate how well this carries to different environments with less organization in the forest structure. 

THe physical site used seems to be such where a backpack campaign is reasonable, since the tree density does not allow vehicle based scanning. Although, an UAV scanning could be more economical (but more difficult to achieve results). 

Response:We greatly appreciate your positive comments about the manuscript. We are honored that our method arouses your interest. The positive comments you made are the greatest encouragement to us. Thanks for pointing out some grammatical errors in our paper, which is very useful for us.

Comments:

ground-based laser scanner point cloud --> ground-based laser scanner point cloud.

Change "radar" to LiDAR (light detection and ranging) or "laser scan" everywhere.

Response:Thank you for your valuable suggestion. I have replaced all occurrences of "radar" with "lidar" throughout the entire text.

  1. 11, 2. Denoising: You should either use boldface for this "subsubsection" name, or make it easy to find (check instructions for aothors) Same with previous topic 1.: "Semantic-Segmentation Editing"

Response:Thank you again for your valuable suggestions. I have now used bold font for all six subheadings starting from page 11 and highlighted them in the text. I have also highlighted lines 377 to 485 on page 10.

  1. 11-12: Reveal the algorithmic structure better in topics 2-5. You don't need to  go to the complete format of DMM module algorithm (p. 15), but present steps in separate lines and make mathematical formula readable. 

Response:Thank you very much for your valuable suggestions. The preprocessing data set steps are described on page 11 of the original text. This part only covers the data processing flow and is not the core algorithm of this paper. I have improved the wording and re-described it from line 377 to line 480 on page 10, highlighting the use of formulas. As for the algorithm on page 15 of the original text, I have gradually described the algorithm structure using formulas, specifically on lines 557 to 583 on page 15 of the new manuscript, and highlighted it in the text.

  1. 13: Table 2: make this one narrow and clear by e.g. giving sensors shorthand symbols in the text and the caption text: V64: Velodyne 64 ...,

: SIC, SICK ... and so on. Change "Points (Millions)" to "Points \\ ($10^6$)

Response:I have made modifications to Table 2, as seen on page 13, row 495, and highlighted it.

  1. 14: Moudule --> Module

Response:Thank you for your suggestions. I apologize for the grammar mistake. I have made the necessary corrections throughout the text.

Introduce L, P and S before the equations, Make sure all variables, 'R', and ones in the Topics 2-5 algorithm descriptions really are types as mathematical variables. Use proper names fro L,P,S everywhere: 

(linearity, planarity, sphericity, see https://www.researchgate.net/publication/311630470_A_Bayesian-Network-Based_Classification_Method_Integrating_Airborne_LiDAR_Data_With_Optical_Images)

Response:Thank you for your valuable suggestions. I have referenced the literature you provided and highlighted it at the end. Additionally, I have defined variables L, S, and P in advance in the original text. The specific modifications can be found on pages 14, lines 534 to 540, with corresponding highlights in the text.

  1. 16: I think usage of P(.): shoukd be like this: 

argmin ... + \lambda \sum_{(i,j)\in ??? } P(N_i \neq N_j) ) 

where P(false)= 0 and P(true) = 1 (introducing a predicate function P)

and \lambda is the regularization weight (there has to be this one?). But if I have understood it wrong, then most of the readers woukd do the same, and then it is better for you to explain it better.

 Response:Thank you very much for your suggestion. In the text, "P" corresponds to λ,Part of the regularization term. This is a problem of finding the optimal energy, for more details please refer to the literature:

Guinard, S., & Landrieu, . “Weakly supervised segmentation-aided classification of urban scenes from 3D LiDAR point clouds.  ISPRS Workshop ,vol.3,pp.1321,2018

In the main text, on page 16, line 582, I provided additional explanations and highlighted them.

4 Discuss --> Discussion

4.1 Evaluate our method --> Method evaluation

5 conclusion --> Conclusion

Response:Thank you very much for your suggestions. I apologize for the mistake that was made at that time. I have already made modifications to the headings of Section 4, 4.1, and Section 5, and highlighted them.

Round 2

Reviewer 2 Report (Previous Reviewer 1)

Comments and Suggestions for Authors

Dear Authors,

   The presented version of the manuscript "PointDMM:A deep learning-based semantic segmentation method for point clouds in complex forest environments" presents several modifications when compared to the second version.

    The writing is more fluent. This makes it easier to understand the entire process required to process lidar point clouds and achieve very promising results.

    I have thoroughly reviewed your manuscript and checked that all my questions have been answered.

    Thank you very much for sending the cover letter.

With best regards,

Comments on the Quality of English Language

Please check the wording of the text as it is still in need of some fine-tuning. Use the ha or m² for unit of area instead of mu.

This manuscript is a resubmission of an earlier submission. The following is a list of the peer review reports and author responses from that submission.

Round 1

Reviewer 1 Report

Comments and Suggestions for Authors

Dear Authors:

  Your article entitled: "PointDMM: A deep learning-based semantic segmentation method for point clouds in complex forest environments" presents a relevant topic regarding the use of terrestrial Lidar data in forestry applications. About your manuscript the wording needs a revision because in some parts the text is very confusing. Watch out for spelling mistakes and use proper punctuation. The study implemented by you used multiple platforms, but from what I could understand, you didn't work with the fusion of data obtained by the different platforms. In chapter 1, it is interesting to highlight the objectives of the study and adequately justify its implementation. In general, the article has serious writing problems and the very structure adopted for the article does not facilitate the understanding of the experiment carried out. The chapter dedicated to the results is very poor and many results (tables, figures, and graphs) are presented without due care in presenting them and the discussion about these results is absent. I submitted a total of 78 comments in the digital file (referring to the revision of the text) and request your attention to the following questions:

1) Abstract: the text is confusing. Please review and present more clearly the problem, the solution developed and the best result obtained. In this case, the 14% increase in accuracy is in relation to which method?

2) Airborne Lidar use ALS: Airborne Laser Scanner

3) Lines 43 to 57: You tried to present a small history of the development of Lidar technology. It would be interesting to chronologically adjust the text covering the gaps mainly from the years 2000 to 2020 where we have important developments with Lidar Multi and hyperspectral and also miniaturized systems for UAV (ULS). Other issues Lidar systems on space platforms (GEDI and Icesat) present a purpose of applications for continental measurements and with much lower precisions when compared to ALS and Lidar systems mounted on mobile platforms.

4) Line 122: forest production. The technology is also used in the sustainable management of natural forests, for example, in the Amazon.

5) Line 129: “the manual fusion parts have unavoidable systematic errors and insufficient fusion accuracy” digital cameras correctly integrated into a Slam-type system tend to produce digital images that can be correctly and automatically fused with the point cloud.

6) It would be interesting to present the metrics of each dataset: number of points, and average density, among others. A figure for each dataset presenting the entire set of points would facilitate the understanding of the locations and also the complexity involved in the processing.

7) In the methodology, please describe the processing steps of the point clouds.

8) To facilitate the replication of the experiments and thereby increase the citations of the article, it is interesting to present the computational resources of hardware and software employed.

9) The presentation of a flowchart with the steps involved in data processing is interesting.

10) Equations 1,2 and 3 describe the terms presented.

11) Equations 4 to 10 describe the terms. Explain equations 7 to 10.

12) Figure 16: Increase the font size to make it easier to read. Detail the operation of the network in the text.

13) Chapter 2.6: detail the sampling stage. Sampling was carried out in what way? Is the table presented in this chapter necessary?

14) It would be interesting to present the original point clouds that generated the trees presented in the results.

15) The results are simply presented without a discussion and even comparison with bibliographical references.

16) In figure 20 the results refer to which dataset and methods?

17) After table 20, a table is presented without proper identification. The values obtained are similar or superior to the works carried out/cited in the bibliographical references.

18) Chapter 3.3: Please review. What's the idea here?

19) Figure 17/21: what does this figure represent? What can be interpreted from the analysis of this figure?

20) Conclusions: conclusions are presented such as time consumption, and superiority of the algorithm among others that are not presented in the results and discussions.

I end my review work by congratulating them for the work done. However, the submitted manuscript does not reflect the necessary quality for its publication.

Respectfully,

Reviewer 2 Report

Comments and Suggestions for Authors

Segmentation of forest PCs is not a settled matter, since methods with practical speed have a rather low segmentation quality or high quality methods are too slow. Besides, the field is plagued by multitude of scanning methods, scanning cicumstances and target environments, so this addition (a DL method based on semantic PC features) may provide incredients to efforts to provide generic, versatile and fast forest PC segmentation.

Comments: Before detailed list, a general recommendation:
  - improve the connection of the text and equations. Introduce 
    the meaning of the equation and variables used either before or right after ( ... (Eq number) newline where ...).
  - check carefully, that variables get introduced in the right order. There seems to be some confusion with some variables (get mentioned much later in the text). 

p. 4: There are 2 topic lists 1-2-3 and 1-2-3 now hidden in the text. Check MDPi instructions how to use lists, and make the list structure (very useful and clarifyuing in this spot) more pronounced typographically. 

p. 5: The end of introduction seems to be incomplete. 

p.9: IoU introduce this "intersection over union". Define also A_IoU

p.9: define cij and L in Eq.2 (apparently the confusion matrix and some sort of a nominal length. Even these might be found form the reference, it is customary that readers are not expected to browse variable definitions form references...

P. 10: Introduce lambda_i's (eigenvalues of a local covariance matrix). Same with f, (+), T, P, alpha, W, socre and \hat{f}. 

p. 13: Completet the paragraph and introdue Q,g, delta

Q(x),to --> substitute ',' with a blanko

function Q studied. where --> function Q is studied next. f is a differentiable ...  R^n to R with a penalty g (?). 

L&O proposed in 2017 -_> a proper reference [xx] , where you bind L&O (and maybe the year, if L&O have other contributions elsewhere). 

Where g is diff... in v --> introduce v. Do you mean: "Our algorthm finds smooth points of f () there, where g is differentiable in v"? Use small f if you mean it. If you refer to Alg 1 F, you have to (again) introduce F (and some other new variables (maybe V in the Alg 1 is the small v in the text). 

Also, introduce the algorithm and the cut pursuit by opening a new paragraph for it (an alg is a big thing, and readers expect a parade introduction for it). 

p. 13: ATSE --> maybe a DMM ATSE module? Show the connection to DMM, if it is so.

p. 15: improve the fst sentence in Ch. 2.5: 

Ni and vi (p.16). Is this v same as shjown earlier (or was that v a V)?
Introduce the bullet product symbol. Introduce a and k. 

p. 16 nvida --> nvidia
The column of a deformed (and partially missing) table starting with "Training hyperparameters": either remove this or completet a table.

p. 17: over-segmented segmented --> maybe remove a word?

p. 20: Fig. 17: Shrurb --> shrub , moudle --> module 

The last paragraph not finished, complete it. 

Reviewer 3 Report

Comments and Suggestions for Authors

Dear Editor,

This work is no longer in good shape. Its text has flaws that prevent me to understand it properly. For instance, the first paragraph has many terms and sentences without clarification, such as “reference data” (reference for what?), “at home and abroad” (abroad from where?), etc. Besides, the second paragraph is almost impossible to follow. It has 80 lines (!) full of inconsistent sentences describing other work, while the Result section has only 10 lines and there is no Discussion section. The authors must rewrite the text, send it for English proofreading, and resubmit it to restart the appreciation.